# YoooP: You Only Optimize One Prototype per Class for Non-Exemplar Incremental Learning

**Jiangtao Kong**                                                  *jkong01@wm.edu*
*Department of Computer Science*
*William & Mary*

**Zhenyu Zong**                                                   *zzong@wm.edu*
*Department of Computer Science*
*William & Mary*

**Tianyi Zhou**                                                   *tianyi@umd.edu*
*Department of Computer Science*
*University of Maryland*

**Huajie Shao**                                                   *hshao@wm.edu*
*Department of Computer Science*
*William & Mary*

**Reviewed on OpenReview:** *https://openreview.net/forum?id=FYe66NLDkO&noteId=XAs1TQUpyQ*

## Abstract

Incremental learning (IL) usually addresses catastrophic forgetting of old tasks when learning new tasks by replaying old tasks' raw data stored in memory, which can be limited by its size and the risk of privacy leakage. Recent non-exemplar IL methods store class centroids as prototypes and perturb them with high-dimensional Gaussian noise to generate synthetic data for replaying. Unfortunately, this approach has two major limitations. First, the boundary between embedding clusters around prototypes of different classes might be unclear, leading to serious catastrophic forgetting. Second, directly applying high-dimensional Gaussian noise produces nearly identical synthetic samples that fail to preserve the true data distribution, ultimately degrading performance. In this paper, we propose YoooP, a novel exemplar-free IL approach that can greatly outperform previous methods by only storing and replaying one prototype per class even without synthetic data replay. Instead of merely storing class centroids, YoooP optimizes each prototype by (1) shifting it to high-density regions within each class using an attentional mean-shift algorithm, and (2) optimizing its cosine similarity with class-specific embeddings to form compact, well-separated clusters. As a result, replaying only the optimized prototypes effectively reduces inter-class interference and maintains clear decision boundaries. Furthermore, we extend YoooP to YoooP+ by synthesizing replay data preserving the angular distribution between each class prototype and the class's real data in history, which cannot be obtained by high-dimensional Gaussian perturbation. YoooP+ effectively stabilizes and further improves YoooP without storing real data. Extensive experiments demonstrate the superiority of YoooP/YoooP+ over non-exemplar baselines in terms of different metrics. The code is released at `https://github.com/Snowball0823/YoooP.git`.

## 1 Introduction

Catastrophic forgetting McCloskey & Cohen (1989) refers to deep neural networks forgetting the acquired knowledge from the previous tasks disastrously while learning the current task. This stands in stark contrast

to human learning, where new knowledge is integrated without erasing prior understanding. To bridge this gap, incremental learning (IL) Gepperth & Hammer (2016); Wu et al. (2019); Douillard et al. (2022); Wang et al. (2022a); Goswami et al. (2024) has emerged as a paradigm that enables AI systems to continuously learn from evolving data.

In the past few years, a variety of methods Roady et al. (2020); Cong et al. (2020); Wang et al. (2021); Xue et al. (2022) have been proposed to mitigate catastrophic forgetting in IL. In this work, we are interested in a very challenging scenario, called class-incremental learning (CIL). CIL is particularly challenging because it requires the model to recognize all learned classes without any task identifier during inference. CIL is especially susceptible to catastrophic forgetting due to overlapping feature representations between old and new tasks Zhu et al. (2021b). To address this issue, many prior studies have adopted *exemplar-based approaches* Rebuffi et al. (2017); Wu et al. (2019); Zhao et al. (2020); Wang et al. (2022b) that store a subset of old class samples in a memory buffer for replay. However, these methods face inherent limitations related to memory capacity and privacy. Thus, *non-exemplar-based methods* Li & Hoiem (2017); Lopez-Paz & Ranzato (2017); Mallya & Lazebnik (2018); Cong et al. (2020); Xue et al. (2022) have been proposed, which avoid storing raw data by relying on regularization, parameter isolation, or generative models to mitigate catastrophic forgetting. Unfortunately, solely applying regularization is often insufficient, parameter isolation increases network size, and generative models can be unstable.

Recently, prototype-based methods Zhu et al. (2021b;a); Petit et al. (2023); Rypeść et al. (2024); Magistri et al. (2024) have attracted attention in non-exemplar CIL. These approaches store a single *class-mean prototype* as the class centroid for each old class and replay synthetic data augmented from these prototypes in future tasks. Notably, PASS Zhu et al. (2021b) augments stored prototypes via high-dimensional Gaussian noise. Several follow-up works, such as EFC Magistri et al. (2024) and AdaGauss Rypeść et al. (2024), also rely on Gaussian-based prototype augmentation by storing class means and covariances to sample synthetic features. However, our observations reveal that such an augmentation strategy may actually degrade prediction accuracy (Sec. 4.1). The underlying issue is twofold. First, simply generating class-mean prototypes without further optimization leads to diffuse clusters of embeddings with unclear decision boundaries between classes (Sec. 4.3). Consequently, the clusters around prototypes are less representative, resulting in serious catastrophic forgetting while training future tasks. Second, as shown in Fig. 1(a)- Prototype Augmentation, directly assuming the class feature distribution can be approximated by Gaussians, and adding high-dimensional Gaussian noise on prototypes yields nearly identical synthetic samples that fail to capture the true distribution of old tasks' class embeddings, and decrease performance.

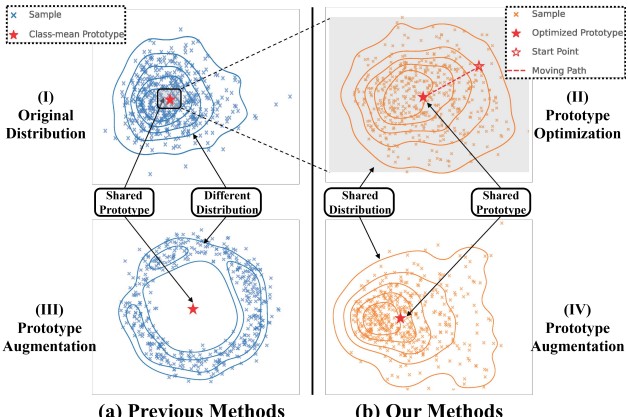

**(a) Previous Methods**   **(b) Our Methods**

Figure 1: Comparison of previous prototype-based (**left**) methods with our YoooP/YoooP+ (**right**). Previous methods (*e.g.*, PASS) typically store and average all class embeddings to form a class-mean prototype, which is then augmented inappropriately (*e.g.*, by directly adding high-dimensional Gaussian noise). In contrast, YoooP adopts a mini-batch attentional mean-shift method, which constructs representative prototypes from a small batch of embeddings. Prototype optimization further refines these prototypes to form a more compact feature space. YoooP+ further enhances performance by synthesizing data with stored original angular distribution of old tasks. Comparing (I) and (II) demonstrates the benefit of prototype optimization in forming compact feature spaces, while comparing (III) and (IV) shows the advantage of our augmentation strategy in preserving realistic sample distributions.

Motivated by these limitations, we propose to optimize prototype learning and develop a novel prototype augmentation strategy for CIL. In this work, we introduce YoooP, a new non-exemplar CIL method that stores and replays only one representative prototype per class without relying on synthetic data. The main challenge is obtaining well-separated prototypes that capture the essence of their respective classes. To achieve this, we introduce prototype optimization (Fig. 1 (II)). First, we employ a mini-batch attentional

mean shift-based method to shift each class prototype toward high-density regions of its corresponding embeddings (the moving path in Fig. 1 (II)). Next, we optimize the angular distance between the prototype and the class-specific embeddings to form tight, compact clusters (comparing the (I) and (II) in Fig. 1). As a result, each prototype becomes highly representative of its class. Consequently, only replaying the prototype can effectively reduce inter-class interference and maintain clear decision boundaries. Building on YoooP, we further develop YoooP+, which extends our approach with a novel prototype augmentation technique. Instead of directly adding high-dimensional Gaussian noise, which implicitly assumes that class feature distributions follow Gaussian patterns, YoooP+ synthesizes data by combining a rotation matrix in high-dimensional space with the stored angular (cosine similarity) distribution between each class prototype and its corresponding real features. As shown in Fig. 1 (IV), this strategy produces synthetic data that more faithfully reflects the true embedding distribution of old classes and yields higher-quality samples.

**Our contributions** are four-fold: 1) We propose YoooP, a novel non-exemplar CIL algorithm that stores and replays a single prototype per class without synthetic data. 2) To the best of our knowledge, we are the first to explore prototype optimization in CIL, which effectively reduces inter-class interference and maintains clear decision boundaries. 3) We extend YoooP to YoooP+, which develops a new prototype augmentation technique that synthesizes high-quality data reflective of the original distribution. 4) Extensive evaluations on multiple benchmarks demonstrate that both YoooP and YoooP+ significantly outperform non-exemplar baselines in terms of average incremental accuracy, average accuracy, and average forgetting.

## 2 Related Work

**Regularization-based method.** This method aims to alleviate catastrophic forgetting by introducing additional regularization terms to correct the gradients and protect the old knowledge learned by the model Li & Hoiem (2017); Rannen et al. (2017); Kirkpatrick et al. (2017); Lee et al. (2017); Liu et al. (2018); Masana et al. (2022). Existing works mainly adopt weight regularization to reduce the impact of learning new knowledge on the weights that are important for old tasks. However, it is very hard to design reasonable and reliable metrics to measure the importance of model parameters. Thus, solely using regularization-based methods is always insufficient.

**Parameters isolation-based method.** This line of work can be divided into dynamic network expansion and static network expansion. Dynamic network expansion methods adopt individual parameters for each task, so they need a large memory to store the extended network for each previous task during training Yoon et al. (2017); Ostapenko et al. (2019); Yan et al. (2021); Xue et al. (2022). Conversely, static network expansion approaches Serra et al. (2018); Mallya & Lazebnik (2018); Mallya et al. (2018); Zhu et al. (2022) dynamically expand the network if its capacity is not large enough for new tasks, and then adapt the expanded parameters into the original network. Those methods can achieve remarkable performance, but they are not applicable to a large number of tasks.

**Data replay-based method.** This solution Wu et al. (2018); Rostami et al. (2019); Cong et al. (2020) mainly employs deep generative models to generate synthetic samples of old classes in order to mitigate privacy leakage. Most existing works Shin et al. (2017); Rios & Itti (2018); Ostapenko et al. (2019); Lesort et al. (2019) focus on Variational Autoencoder (VAE) and Generative Adversarial Network (GAN). However, these methods suffer from the instability of generative models and inefficient training for complex datasets.

**Prototype-based method.** Recent works Zhu et al. (2021b;a); Petit et al. (2023); Goswami et al. (2023); Rypeść et al. (2024); Magistri et al. (2024) avoid generating pseudo samples by storing class-representative prototypes and adapting prototype-based strategies to mitigate catastrophic forgetting. A key line of work, including PASS Zhu et al. (2021b), IL2A Zhu et al. (2021a), FeTrIL Petit et al. (2023), AdaGauss Rypeść et al. (2024), and EFC Magistri et al. (2024), uses these prototypes to generate synthetic samples for replay-based training. PASS directly adds high-dimensional Gaussian noise to stored prototypes, while AdaGauss and EFC incorporate adaptive covariance modeling or elastic attraction to improve sample realism. However, all these methods rely on unoptimized class-mean centroids and Gaussian assumptions, which may lead to unrealistic or overlapping synthetic samples that poorly capture class semantics, especially in high-dimensional feature spaces. In contrast, FeCAM Goswami et al. (2023) does not generate synthetic data, but instead uses a Mahalanobis-distance-based classifier built upon saved means and covariances. Yet, this method freezes the

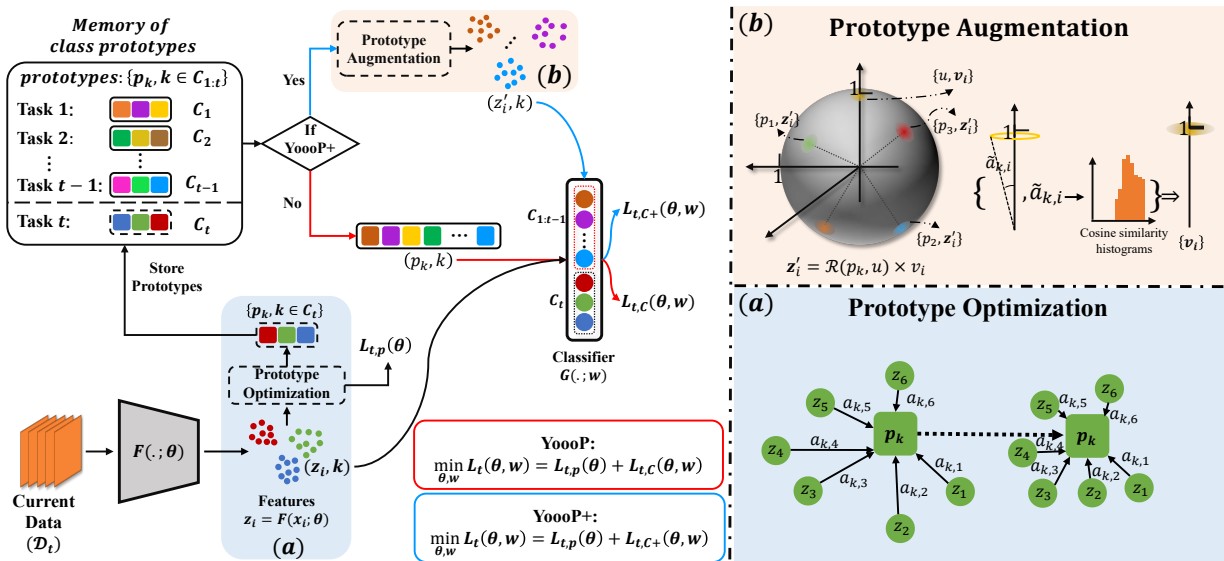

Figure 2: Framework of the proposed YoooP and YoooP+. YoooP only needs to replay one stored prototype for each class while YoooP+ is trained on synthetic data generated from stored prototypes. (a) Prototype optimization aims to learn a compact feature space and obtain a representative prototype for each class. (b) Prototype augmentation aims to generate synthetic data of old classes from stored prototypes and angular distribution using a $m$-dimensional space rotation matrix.

feature extractor after the first task and thus struggles under large task shifts or limited prior knowledge, due to its inability to adapt learned features. Another recent study Goswami et al. (2024) has leveraged class-mean prototypes as classifiers, using a Nearest Class Mean (NCM) approach to correct prototype drift during sequential training. Nevertheless, compared to standard gradient-based classifiers, NCM is limited in its ability to learn complex decision boundaries when task distributions shift significantly. In this work, we propose prototype optimization during training to obtain compact and discriminative clusters. On top of that, we introduce an angular-based prototype augmentation strategy that avoids unrealistic sampling in Euclidean space, thereby generating high-quality synthetic data and effectively mitigating catastrophic forgetting.

## 3 Proposed Method

In this section, we first describe YoooP, which optimizes the prototype for each class using an attentional mean-shift method. To further improve prediction accuracy, we extend YoooP to YoooP+ by generating synthetic data from the stored prototypes.

**Problem Description.** Given a sequence of tasks, each associated with a set of classes $C_t$ and a training set $\mathcal{D}t \triangleq (x_i, y_i)i = 1^{n_t}$ with $y_i \in C_t$, class-incremental learning (CIL) aims to train a model $f(x; [\theta, w]) \triangleq G(F(x; \theta); w)$ that predicts probabilities for all classes $C_{1:t} \triangleq \bigcup_{i=1}^{t} C_i$ without catastrophic forgetting. The model consists of a feature extractor $F(\cdot; \theta)$, which produces compact representations, and a classifier $G(\cdot; w)$, with predictions given by softmax$(G(F(x; \theta); w))$. Since the parameters $\theta$ and $w$ are updated solely using the current task's data, the model typically suffers from catastrophic forgetting. To address this challenge, we propose YoooP and its extension YoooP+ to preserve clear class boundaries as tasks are learned sequentially.

### 3.1 YoooP

YoooP comprises two main components: (i) prototype optimization to learn a compact feature space and obtain a representative prototype for each class, and (ii) new task learning with prototype replay.

For (i), we propose a mini-batch attentional mean-shift-based method to shift each class prototype toward high-density regions of its embeddings. Then, we optimize the angular distance between the prototype and the class-specific embeddings to form a tight, compact cluster. As a result, each prototype becomes highly representative. Consequently, replaying only the optimized prototype effectively reduces inter-class interference and maintains clear decision boundaries.

For (ii), when learning a new task, we augment its training set with the stored prototypes from previous tasks. In the YoooP, replaying only the memorized prototypes of old classes can efficiently retain clear boundaries between classes and mitigate catastrophic forgetting.

### 3.1.1 Prototype Optimization

To obtain a representative prototype *without storing all sample embeddings* (unlike the traditional class-mean prototype), we propose a mini-batch attentional mean-shift-based method. Specifically, for task-$t$ with classes $C_t$, and for each class $k \in C_t$, we construct a graph of sample representations $z_i = F(x_i; \theta)$ connected to the prototype $p_k$. We then shift $p_k$ toward a high-density region by moving it toward a weighted average of the normalized representations of all samples in class-$k$ and subsequently normalizing $p_k$, *i.e.*,

$$p_k \leftarrow (1 - \lambda)p_k + \lambda \sum_{i \in [n_t]: y_i = k} a_{k,i} \cdot \frac{z_i}{\|z_i\|_2}, \quad p_k \leftarrow \frac{p_k}{\|p_k\|_2}, \tag{1}$$

where $\lambda$ controls the step size of the mean-shift and $n_t$ is the size of the training set for task-$t$. Unlike the original mean-shift algorithm, the weights $a_{k,i}$ are determined by learnable dot-product attention between each sample $z_i$ and the prototype $p_k$, *i.e.*,

$$a_k \triangleq \text{softmax}(\bar{a}_k), \quad \bar{a}_k \triangleq [\bar{a}_{k,1}, \cdots, \bar{a}_{k,n_t}], \quad \bar{a}_{k,i} = c(z_i, p_k) \triangleq \frac{\langle z_i, p_k \rangle}{\|z_i\|_2 \cdot \|p_k\|_2}. \tag{2}$$

In practice, when $n_t$ is large, we apply a *mini-batch version* of Eq. 1 over multiple steps, replacing $i \in [n_t]$ with $i \in B$, where $B$ is a mini-batch.

After obtaining a representative prototype for each class, we optimize the similarity between class-specific embeddings and their prototype, to form a compact cluster of each class in the feature space. To achieve that, We train the representation model $F(\cdot; \theta)$ to produce $z_i = F(x_i; \theta)$ such that each sample is close to its class prototype and distant from other classes' prototypes. This is achieved by minimizing the loss as follows,

$$L_{t,P}(\theta) \triangleq \frac{1}{|C_t|} \frac{1}{n_t} \sum_{k \in C_t} \sum_{i \in [n_t]: y_i = k} \ell \left( [c(z_i, p_j)]_{j \in C_{1:t}}, k \right), \tag{3}$$

where $\ell(\cdot, \cdot)$ is a cross-entropy loss, in which the first argument is the logits defined by a similarity measure $c(\cdot, \cdot)$, and the second argument $k$ is the ground truth label. Optimizing this loss ensures that embeddings concentrate around their respective prototype, forming compact clusters. Thus, replaying only one prototype per class effectively reduces harmful inter-class interference while training future tasks.

### 3.1.2 New Task Learning with Prototype Replay

To mitigate catastrophic forgetting of previous tasks' classes $C_{1:t-1}$, YoooP replays stored class prototypes during new task training. Specifically, we augment the training set for task $t$ with prototypes from previous classes $C_{1:t-1}$. The classification objective for all classes $C_{1:t}$ is,

$$L_{t,C}(\theta, w) \triangleq \frac{1}{|C_t| \cdot n_t} \sum_{k \in C_t} \sum_{i \in [n_t]: y_i = k} \ell([c(z_i, w_j)]_{j \in C_{1:t}}, k) + \frac{1}{|C_{1:t-1}|} \sum_{k \in C_{1:t-1}} \ell([c(p_k, w_j)]_{j \in C_{1:t}}, k), \tag{4}$$

Furthermore, to preserve the performance of learned prototypes while training future tasks, it is necessary to retain the performance of extractor $F(\cdot; \theta)$ on previous tasks. Following previous work Hou et al. (2019); Zhu et al. (2021b), we employ knowledge distillation (KD) Hou et al. (2019) as follows,

$$L_{t,KD}(\theta) \triangleq \frac{1}{n_t} \sum_{i \in [n_t]} \|F(x_i; \theta) - F(x_i; \theta_{t-1})\|_2^2. \tag{5}$$

Thus, the overall training objective at task $t$ is,

$$\text{YoooP}: \quad \min_{\theta,w} L_t(\theta,w) = L_{t,P}(\theta) + L_{t,C}(\theta,w) + \gamma * L_{t,KD}(\theta). \quad (6)$$

where $\gamma$ is the weight of KD loss.

In summary, $L_{t,P}(\theta)$ shifts the current task's embeddings toward their class prototype, forming compact clusters. $L_{t,C}(\theta,w)$ trains the model using both current task data and the replayed prototypes, while $L_{t,KD}(\theta)$ preserves the extractor's performance on previous tasks. Together, these objectives enable the model to learn new tasks without forgetting previous ones.

## 3.2 YoooP+

Although prototype-only replay in YoooP is highly effective in mitigating catastrophic forgetting, it is still insufficient to reflect the true embedding distribution of old classes without replaying raw instances. Hence, we propose an extension, YoooP+, which replays synthetic data augmented from the stored prototypes.

### 3.2.1 Prototype Augmentation.

To generate high-quality synthetic data that matches the real embedding distribution of old classes, we propose a novel prototype augmentation strategy. Our method draws synthetic data for each class from the real angular distribution between the class prototype and its specific embeddings. To simplify the augmentation, we first normalize each prototype to a unit vector, generate synthetic data in the normalized space, and then rotate the synthetic data back. As shown in Fig. 3,

we sample cosine similarity values from the stored real angular distribution, $P(\bar{a}_{k,i})$, which is represented by a histogram with $N_b$ bins. These sampled cosine similarities are used to generate synthetic data for each class. Consequently, the angular distribution between each class prototype and its synthetic data faithfully preserves $P(\bar{a}_{k,i})$. In contrast, approaches like PASS add high-dimensional noise to saved prototypes, causing significant divergence from the actual angular distribution.

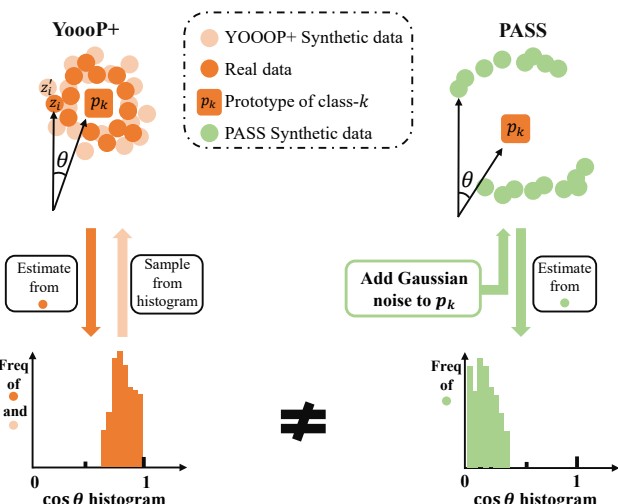

Specifically, by using the stored $P(\bar{a}_{k,i})$, we are able to synthesize a data point $z_i'$ that has a similar angular distance to the prototype $p_k$ as $z_i$ for replaying. This leads to YoooP+ whose replay of each previous class is conducted on multiple synthetic data points instead of a single prototype.

Figure 3: Synthetic data distributions of YoooP+ and PASS with prototype augmentation: **YoooP+ preserves the original angular distribution by a histogram with $N_b$ bins.**

In particular, **we firstly derive a rotation matrix** $\mathcal{R}(p_k, \boldsymbol{u})$ that can recover $p_k$ from a unit vector $\boldsymbol{u} = [1, 0, \cdots, 0]$ on an unit $m$-sphere, *i.e.*, $p_k = \mathcal{R}(p_k, \boldsymbol{u}) \times \boldsymbol{u}$. To synthesize a sample $z_i'$ of class-$k$ as a proxy to $z_i$ (a previously learned sample of class-$k$), **we then randomly draw $\boldsymbol{v_i}$** in the vicinity of $\boldsymbol{u}$, *i.e.*,

$$\boldsymbol{v_i} = [\tilde{a}_{k,i}, \epsilon_2, \cdots, \epsilon_m], \quad \tilde{a}_{k,i} \sim P(\bar{a}_{k,i}) \quad (7)$$

To ensure $\|\boldsymbol{v_i}\|_2 = 1$, we draw $\epsilon_i \sim \mathcal{N}(0,1)$ for $i \in \{2, \cdots, m\}$ at first and then rescale them by $\epsilon_i \leftarrow \sqrt{1-(\tilde{a}_{k,i}+\epsilon_1)^2/\sum_{i=2}^m \epsilon_i^2} \cdot \epsilon_i$. Thereby, we have $\boldsymbol{u}^T \boldsymbol{v_i} = \tilde{a}_{k,i}$, whose distribution approximates the distribution of cosine similarity $\bar{a}_{k,i}$ between real sample $z_i$ and its associated class prototype $p_k$.

**Next, we create $z_i'$ from $\boldsymbol{v_i}$.** As $p_k = \mathcal{R}(p_k, \boldsymbol{u}) \times \boldsymbol{u}$, we apply the same rotation matrix $\mathcal{R}(p_k, \boldsymbol{u})$ to $\boldsymbol{v_i}$ to obtain $z_i'$, *i.e.*,

$$z_i' = \mathcal{R}(p_k, \boldsymbol{u}) \times \boldsymbol{v_i}. \quad (8)$$

This operation preserves the similarity between $\boldsymbol{u}$ and $\boldsymbol{v_i}$ in the transformed space between $p_k$ and $z_i'$. By sampling a synthetic data point $z_i'$ for each removed real sample $z_i$, we construct a replay dataset for all seen classes $C_{1:t-1}$.

### 3.2.2 New Task Learning with Prototype Augmentation

After generating the synthetic data, YoooP+ learns a new task task-$t$ by replaying the synthetic dataset $\mathcal{D}_t'$ as follows,

$$\mathcal{D}_t' \triangleq \{(z_i', k) : k \in C_{1:t-1}, z_i' = \mathcal{R}(p_k, \boldsymbol{u}) \times \boldsymbol{v_i}, \boldsymbol{v_i} = [\tilde{a}_{k,i}, \epsilon_2, \cdots, \epsilon_m]\}. \tag{9}$$

The training objective for task-$t$ with replayed synthetic data is

$$L_{t,C+}(\theta, w) \triangleq \frac{1}{|C_t| \cdot n_t} \sum_{k \in C_t} \sum_{i \in [n_t]: y_i = k} \ell(c(z_i, w), k) + \frac{1}{|\mathcal{D}_t'|} \sum_{(z,k) \in \mathcal{D}_t'} \ell(c(z, w), k). \tag{10}$$

Overall, the training objective $L_t(\theta, w)$ of YoooP+ at task-$t$ combines the prototype-learning loss, the synthetic-data replay augmented loss, and the KD loss as follows,

$$\text{YoooP+}: \quad \min_{\theta, w} L_t(\theta, w) = L_{t,P}(\theta) + L_{t,C+}(\theta, w) + \gamma * L_{t,KD}(\theta). \tag{11}$$

### 3.3 Practical Improvement to YoooP/YoooP+

While the KD loss constrains the feature extractor to maintain representational consistency with the previous model, it only applies to current-task data. As a result, it cannot fully prevent parameter drift in the absence of old-task examples. To complement this, we adopt two lightweight yet effective techniques at the parameter level to further mitigate forgetting.

**Model Interpolation.** We apply model interpolation to retain the knowledge of the previous model $\theta_{t-1}$ and avoid overfitting to the current task. Specifically, after learning task-$t$, we update the current $\theta_t$ by the following interpolation between $\theta_{t-1}$ and $\theta_t$, *i.e.*,

$$\theta_t \leftarrow (1 - \beta)\theta_{t-1} + \beta\theta_t, \tag{12}$$

where $\beta \in [0, 1]$ and we set $\beta = 0.6$ in experiments. Since $\theta_t$ is mainly trained on task-$t$, such simple interpolation between $\theta_{t-1}$ and $\theta_t$ leads to a more balanced performance on all tasks.

**"Partial Freezing" of Classifier.** Following Li & Hoiem (2017), instead of completely freezing the classifier parameters for previously learned classes, $w_k, k \in C_{1:t-1}$, we scale down the gradients of these parameters by a small factor $\alpha$. Specifically, the gradient update is modified as follows,

$$\nabla_{w_k} L_t(\theta, w) \leftarrow \alpha \nabla_{w_k} L_t(\theta, w), \quad \forall k \in C_{1:t-1} \tag{13}$$

This strategy helps prevent significant drift of the classifier parameters for previously learned classes.

We provide the complete procedure of YoooP and YoooP+ in Algorithm 1 in Appendix A.

## 4 Experiment

In this section, we first compare the proposed YoooP and YoooP+ with non-exemplar-based baselines in three datasets. Then we assess the quality of synthetic data augmented from memorized prototypes. Lastly, we do ablation studies to explore the impact of different key components. We also explore the sensitivity of hyper-parameters in Appendix B.

**Datasets.** To better evaluate the performance of our proposed methods, we perform on three different scale datasets: CIFAR-100 Krizhevsky et al. (2009), TinyImageNet Yao & Miller (2015), and a subset of ImageNet-1000 Russakovsky et al. (2015) (Sub-ImageNet). CIFAR-100 contains 100 classes of images, which include 50,000 training images and 10,000 test images, and the image size is $32 \times 32$. TinyImageNet contains

200 classes of images, which include 100,000 training images and 10,000 test images, and the image size is $64 \times 64$. For the Sub-ImageNet, the detailed description and the results are shown in Appendix C.

**Experimental settings.** We implement all experiments using PyTorch Paszke (2019), and compare our methods with baselines provided by PyCIL Zhou et al. (2021), a popular toolbox for continual incremental learning (CIL). Following prior works Zhu et al. (2021b;a), we adopt ResNet-18 He et al. (2016) as the backbone network for all methods. For YoooP and YoooP+, we train with the SGD optimizer with an initial learning rate of 0.01, which is decayed by a factor of 0.1 every 20 epochs. Models are trained for 60 epochs per task using a batch size of $B = 256$. For knowledge distillation loss (Eq. 6 for YoooP and Eq. 11 for YoooP+), we set a large weight $\gamma = 30$. Additionally, we save the cosine similarity distribution into a histogram consisting of $N_b = 100$ bins within the interval $[0, 1]$ for prototype analysis (Fig. 3). For evaluation, we consider two standard CIL settings: "zero-base," where the total classes are evenly split into multiple incremental phases (*i.e.* 5 or 10 phases) trained sequentially, and "half-base," where half of the classes are learned initially and remaining classes are evenly incremented. Following previous practice Zhu et al. (2021b), all classes in each dataset are arranged in a fixed random order, and reported results are averaged across three runs with a fixed random seed for reproducibility. Additionally, we also perform 20 phases under different settings on TinyImageNet in Appendix D.

**Baselines.** We compare our proposed YoooP and YoooP+ methods with representative non-exemplar-based methods, including LwF Li & Hoiem (2017), PASS Zhu et al. (2021b), SSRE Zhu et al. (2022), IL2A Zhu et al. (2021a), FeTrIL Petit et al. (2023), FeCAM Goswami et al. (2023), EFC Magistri et al. (2024), AdaGauss Rypeść et al. (2024), DS-AL Zhuang et al. (2024) and ADC Goswami et al. (2024). Notably, ADC and FeCAM employ an NCM classifier Rebuffi et al. (2017) while YoooP/YoooP+ and other baseline methods perform with a normal classifier with softmax. To further show the impressive performance of our proposed methods, we also compare the performance with some exemplar-based methods on TinyImageNet in Appendix E.

**Protocol.** We evaluate all methods using three commonly adopted metrics in IL: average incremental accuracy Rebuffi et al. (2017) (AIA), the average accuracy after training the last task Rebuffi et al. (2017) (AA), and average forgetting Chaudhry et al. (2018) (AF).

## 4.1 Main Results

In this section, we compare the performance of our proposed methods YoooP and YoooP+ across various datasets under different settings with various baselines and discuss the results.

**Evaluation on zero-base setting.** Next, we evaluate the proposed methods against baseline approaches on CIFAR-100 and TinyImageNet datasets under zero-base settings with 5 and 10 phases. As shown in Tab. 1, our proposed YoooP consistently outperforms non-NCM baselines that rely on the standard classifier layer. Specifically, compared to PASS, which achieves the best results among non-NCM baselines, YoooP achieves accuracy improvements of 4.91% and 7.05% (AIA) and 8.47% and 8.97% (AA) under 5 and 10 phases, respectively, on CIFAR-100. On TinyImageNet, the advantages of YoooP are even more pronounced, exceeding PASS by 13.15% and 13.58% (AIA), and by 17.99% and 13.95% (AA) under the respective phases. This impressive performance is because YoooP leverages prototype optimization, enabling the model to learn more discriminative and compact representations. To further mitigate catastrophic forgetting, YoooP+ augments the training with synthetic data derived from these optimized prototypes. Consequently, YoooP+ surpasses PASS by 9.28%, 11.36% (AIA), and by 13.95%, 13.79% (AA) on CIFAR-100. On TinyImageNet, the performance gap is even more notable: 14.81%, 16.32% for AIA, and 20.27%, 17.94% for AA under the respective phases. Additionally, YoooP surpasses ADC and FeCAM, which employ the Nearest Class Mean (NCM) classifier, on TinyImageNet under zero-base settings, and YoooP+ significantly surpasses ADC on both datasets under zero-base settings. We also show the performance while training the model incrementally on each task in the upper row in Fig. 4.

**Evaluation on half-base setting.** We first evaluate the proposed methods under half-base settings with 5 and 10 phases. Results shown in Tab. 2 indicate that YoooP/YoooP+ consistently performs well across CIFAR-100 and TinyImageNet datasets. On CIFAR-100, YoooP+ achieves performance comparable to FeTrIL, DS-AL, and FeCAM, which attain strong results by training the model only on the

Table 1: Average incremental accuracy (AIA) and average accuracy after training the last task (AA) of the proposed YoooP/YoooP+ and baselines on CIFAR-100 and TinyImageNet under zero-base setting with different phases."b0-10" means zero-base with 10 phases, "b0-5" means zero-base with 5 phases. (*) denotes methods using "Nearest Class Mean" (NCM) classifier. **Bold**: the best among non-exemplar methods. Underline: the second best among non-exemplar methods.

| Datasets | CIFAR-100 | | | | TinyImageNet | | | |
|---|---|---|---|---|---|---|---|---|
| | AIA [%]↑ | | AA [%]↑ | | AIA [%]↑ | | AA [%]↑ | |
| Method | b0-5 | b0-10 | b0-5 | b0-10 | b0-5 | b0-10 | b0-5 | b0-10 |
| LwF Li & Hoiem (2017) | 58.95 | 47.73 | 39.87 | 25.88 | 46.44 | 35.00 | 29.50 | 17.80 |
| SSRE Zhu et al. (2022) | 58.05 | 46.58 | 40.99 | 29.75 | 47.13 | 38.54 | 29.90 | 22.78 |
| IL2A Zhu et al. (2021a) | 59.91 | 42.92 | 43.69 | 27.21 | 42.16 | 33.72 | 25.67 | 22.46 |
| FeTrIL Petit et al. (2023) | 61.41 | 48.61 | 44.54 | 31.07 | 44.55 | 36.51 | 28.23 | 20.59 |
| DS-AL Zhuang et al. (2024) | 66.79 | 56.74 | 50.30 | 42.76 | 49.62 | 47.24 | 36.89 | 33.71 |
| EFC Magistri et al. (2024) | 67.52 | 59.96 | 50.47 | 45.24 | 54.12 | 48.03 | 36.92 | 34.12 |
| AdaGauss Rypeść et al. (2024) | 68.42 | 59.13 | 56.02 | 46.32 | 55.58 | 50.51 | 41.33 | 36.52 |
| *FeCAM Goswami et al. (2023) | 61.27 | 48.93 | 45.78 | 32.11 | 54.37 | 46.03 | 39.96 | 32.02 |
| *ADC Goswami et al. (2024) | 68.60 | 61.93 | 57.49 | 47.00 | 60.09 | 53.51 | 47.46 | 37.32 |
| PASS Zhu et al. (2021b) | 60.33 | 51.94 | 43.61 | 35.81 | 47.11 | 40.15 | 31.05 | 26.31 |
| PASS w/o Aug | 59.02 | 55.47 | 45.19 | 42.41 | 50.25 | 42.71 | 36.27 | 30.17 |
| YoooP (Ours) | 65.24 | 58.99 | 52.08 | 44.78 | 60.26 | 53.73 | 49.04 | 40.26 |
| YoooP+ (Ours) | **69.61** | **63.30** | **57.56** | **49.60** | **61.92** | **56.47** | **51.32** | **44.25** |

Table 2: Average incremental accuracy (AIA) and average accuracy after training the last task (AA) of the proposed YoooP/YoooP+ and baselines on CIFAR-100 and TinyImageNet under half-base setting with different phases."half-10" means half-base with 10 phases, "half-5" means half-base with 5 phases. (*) denotes methods using "Nearest Class Mean" (NCM) classifier. **Bold**: the best among non-exemplar methods. Underline: the second best among non-exemplar methods.

| Datasets | CIFAR-100 | | | | TinyImageNet | | | |
|---|---|---|---|---|---|---|---|---|
| | AIA [%]↑ | | AA [%]↑ | | AIA [%]↑ | | AA [%]↑ | |
| Method | half-5 | half-10 | half-5 | half-10 | half-5 | half-10 | half-5 | half-10 |
| LwF Li & Hoiem (2017) | 49.00 | 37.46 | 27.93 | 17.09 | 34.46 | 23.34 | 21.34 | 13.56 |
| SSRE Zhu et al. (2022) | 63.38 | 61.29 | 52.35 | 49.27 | 50.45 | 47.88 | 40.48 | 39.47 |
| IL2A Zhu et al. (2021a) | 62.97 | 52.44 | 48.89 | 33.08 | 45.27 | 43.34 | 34.54 | 33.73 |
| FeTrIL Petit et al. (2023) | 67.57 | 67.43 | 58.38 | 57.45 | 51.79 | 50.10 | 42.51 | 41.76 |
| DS-AL Zhuang et al. (2024) | **68.42** | **68.37** | **61.35** | **61.23** | 58.41 | 58.08 | 50.92 | 48.86 |
| EFC Magistri et al. (2024) | 67.93 | 64.88 | 56.55 | 54.83 | 56.64 | 53.18 | 44.67 | 40.35 |
| AdaGauss Rypeść et al. (2024) | 65.07 | 62.35 | 58.14 | 54.82 | 58.33 | 56.00 | 50.09 | 47.90 |
| *FeCAM Goswami et al. (2023) | 67.17 | 67.01 | 60.21 | 59.93 | 60.25 | 60.02 | 50.64 | 50.64 |
| *ADC Goswami et al. (2024) | 65.62 | 61.71 | 54.31 | 49.12 | 52.36 | 47.12 | 43.00 | 37.18 |
| PASS Zhu et al. (2021b) | 64.10 | 57.41 | 54.80 | 45.71 | 48.61 | 39.99 | 38.69 | 30.23 |
| PASS w/o Aug | 59.34 | 55.41 | 49.64 | 43.02 | 46.06 | 42.84 | 37.12 | 33.56 |
| YoooP (Ours) | 66.19 | 58.99 | 56.61 | 47.09 | 62.30 | 58.66 | 53.94 | 49.40 |
| YoooP+ (Ours) | 67.40 | 61.83 | 58.54 | 50.81 | **65.57** | **61.72** | **58.56** | **52.40** |

initial task while keeping it fixed for future tasks. Although FeTrIL, DS-AL, and FeCAM effectively preserve initial task knowledge, those strategy severely restricts the model's capacity to adapt and improve representations for subsequent tasks, particularly when these differ significantly from the initial task.

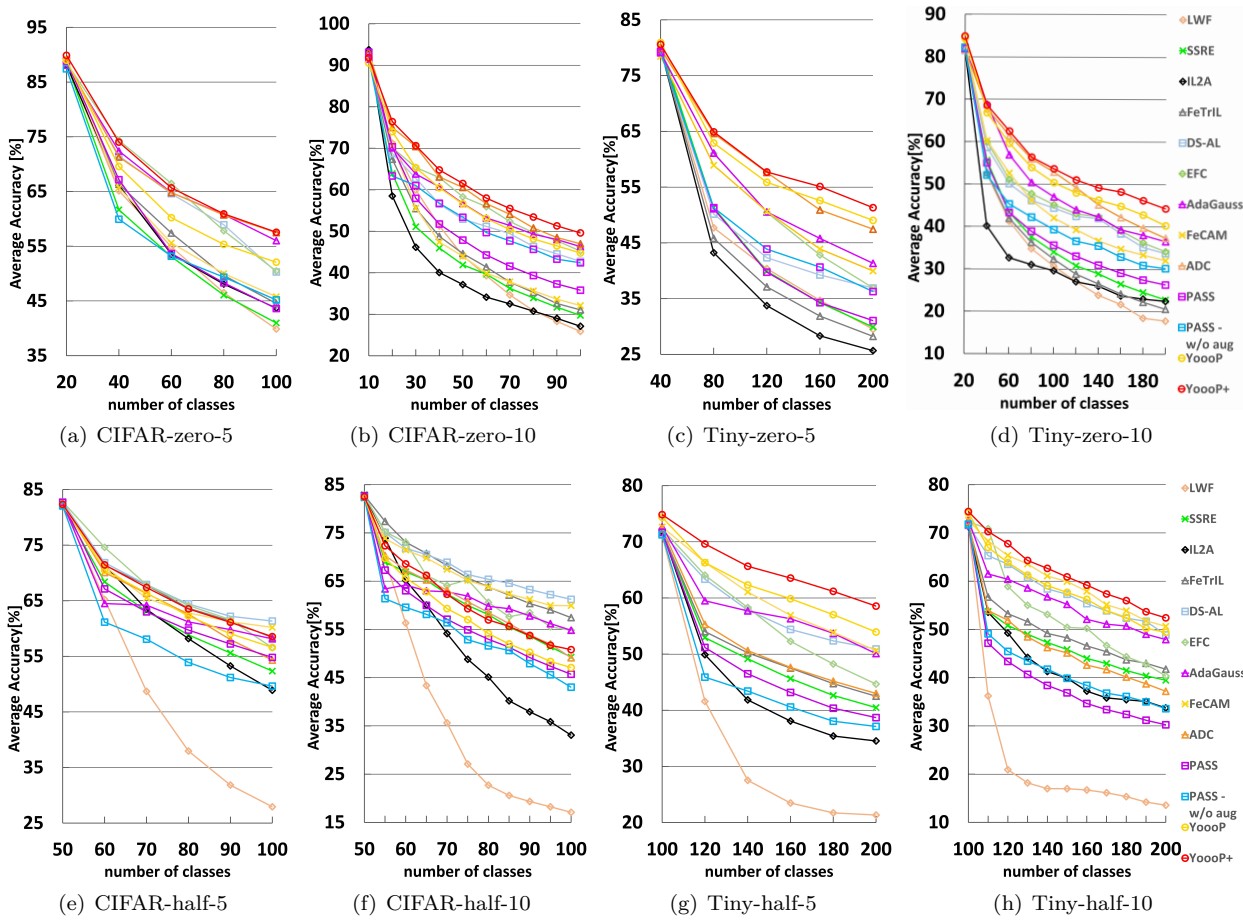

Figure 4: Performance comparison of each task across different methods on CIFAR-100 and TinyImageNet under different settings while training model incrementally. "zero-5,10": "zero-base" with 5 and 10 phases settings. "half-5,10": "half-base" with 5 and 10 phases settings.

Thus, the FeTrIL, DS-AL, and FeCAM fail to achieve great performance if the initial task contains little knowledge and the future tasks have a large gap from the first task. Therefore, the proposed YoooP and YoooP+ beat the FeTrIL, DS-AL, and FeCAM on both datasets under zero-base settings, and YoooP+ surpasses those three methods on TinyImageNet under half-base settings. Specifically, on TinyImageNet, YoooP outperforms DS-AL by 3.89% and 0.58% (AIA), and 3.02% and 0.54% (AA) under the respective phases. Furthermore, YoooP+ exhibits even greater improvements over DS-AL, achieving 7.16% and 3.64% (AIA), and 7.64% and 3.54% (AA). YoooP and YoooP+ also clearly surpass ADC by a large margin, reinforcing our methods' capability to dynamically capture richer class representations, particularly on more challenging datasets like Tiny-ImageNet. The incremental accuracy progression under the half-base setting is depicted in the bottom row of Fig. 4.

Table 3: Average forgetting (AF) of the proposed YoooP/YoooP+ and baselines on CIFAR-100 and Tiny-ImageNet under zero-base setting with different phases. "b0-10" means zero-base with 10 phases, "b0-5" means zero-base with 5 phases. (⋆) denotes methods using "Nearest Class Mean" (NCM) classifier. **Bold**: the best among non-exemplar methods. Underline: the second best among non-exemplar methods.

| AF [%]↓ | CIFAR-100 | | TinyImageNet | |
|---|---|---|---|---|
| Methods | b0-5 | b0-10 | b0-5 | b0-10 |
| LwF Li & Hoiem (2017) | 43.80 | 51.80 | 45.79 | 54.40 |
| SSRE Zhu et al. (2022) | **15.44** | **12.13** | 16.31 | 19.94 |
| IL2A Zhu et al. (2021a) | 26.94 | 25.07 | 20.89 | 26.10 |
| FeTrIL Petit et al. (2023) | 18.88 | 16.14 | 15.13 | **15.32** |
| ⋆ADC Goswami et al. (2024) | 19.15 | 21.21 | 15.35 | 23.92 |
| PASS Zhu et al. (2021b) | 23.66 | 18.78 | 22.00 | 20.69 |
| PASS w/o Aug | 28.11 | 29.55 | 24.01 | 26.00 |
| YoooP (Ours) | 21.24 | 22.69 | 16.49 | 25.02 |
| YoooP+ (Ours) | 18.75 | 15.41 | **14.28** | 19.24 |

**Evaluation on Average Forgetting.** We also evaluate the average forgetting (AF) under zero-base settings on CIFAR-100 and TinyImageNet. As shown in Tab. 3, our proposed methods, YoooP and YoooP+, achieve

relatively low average forgetting compared to most baseline methods. Although some baselines exhibit lower AF, this is primarily due to their sharp accuracy decline in initial incremental phases followed by persistently low performance, as shown in Tab. 1 and clearly illustrated by the incremental accuracy curves in the bottom row of Fig. 4. In contrast, YoooP and YoooP+ effectively maintain higher incremental accuracy (AIA and AA) throughout all phases (Tab. 1), genuinely mitigating catastrophic forgetting while maintaining superior overall performance.

**Discussion.** Our experimental results yield three main findings: (1). Comparing PASS w/o Aug and PASS in Tab. 1 and Tab. 2, directly adding high-dimensional Gaussian noise for prototype augmentation can hurt performance, especially under zero-base settings and the half-base setting with 10 phases on TinyImageNet. This is because high-dimensional Gaussian noise produces sparse, nearly equidistant data (the curse of dimensionality). In contrast, YoooP+ generates synthetic data in angular space using a rotation matrix while preserving the real angular distribution, which consistently improves prototype performance. (2). The ADC with NCM shows an advantage only in the CIFAR-100 zero-base setting. As shown in Tab. 1 and Tab. 2, ADC outperforms YoooP on CIFAR-100 under zero-base conditions but does not surpass YoooP+ and fails to outperform YoooP in other settings. Furthermore, in the half-base configuration, several non-NCM methods outperform ADC. This occurs because NCM relies solely on class-mean prototypes and avoids gradient updates to mitigate catastrophic forgetting, limiting its ability to learn complex decision boundaries when task distributions shift significantly. In contrast, a normal classifier with softmax benefits from gradient updates that enable the learning of complex boundaries and better generalization. In the half-base setting, where the initial task has abundant data and later tasks have fewer samples, the pronounced distribution shift leads to a performance drop (comparing the performance of ADC between the two tables) and the loss of ADC's initial advantage. (3). The proposed YoooP and YoooP+ consistently outperform other non-exemplar-based methods across different datasets under various settings. They achieve higher AIA and AA while maintaining lower AF, demonstrating robust performance.

## 4.2 Comparison of Synthetic Data for YoooP+ and PASS

In this experiment, we randomly selected five classes from the CIFAR-100 dataset (task $t$) under a zero-base setting with 10 phases. We then compared the angular distributions of synthetic data generated from stored prototypes by YoooP+ and PASS, as illustrated in Fig. 5. The upper row of Fig. 5 pertains to the original distribution (also the angular distribution stored in YoooP+), representing the cosine similarities between the representations $F(\cdot; \theta)$ and the stored prototypes for each class. In contrast, the bottom row of Fig. 5 shows the distributions of cosine similarities between the same prototypes and the synthetic data generated by PASS. We observe that PASS produces synthetic samples whose cosine similarities are heavily concentrated near 1.0, indicating that these augmented data are nearly identical to the stored prototypes. This narrow distribution leads to reduced diversity and, consequently, diminished performance, as also evidenced in Fig. 4, Tab. 1, and Tab. 2. In comparison, YoooP+ draws its synthetic data based on the original cosine-similarity distribution (top row). This strategy more effectively restores the representations of the original data and helps YoooP+ generate higher-quality synthetic samples than PASS.

## 4.3 Comparison of the representations for YoooP+ and PASS

In addition, we compare the learned representations produced by YoooP+ and PASS. Fig. 6 shows the distribution of representations on CIFAR-100 under zero-base setting with 10 phases for the first three tasks. Because YoooP+ employs prototype optimization in the first task, it forms more compact class clusters than PASS. In Tasks 2 and 3, YoooP+ continues to encode input data within well-defined boundaries, whereas PASS does not. In Fig. 6 (b), (c), (e), and (f), the light gray points represent data from previous tasks. We observe in (b) and (c) that YoooP+ keeps old and new tasks well separated, while PASS struggles to distinguish between the distributions of old and current tasks. This is because YoooP+ not only creates compact clusters via prototype optimization but also synthesizes high-quality data from the original cosine-similarity distribution, effectively preserving boundaries for old tasks.

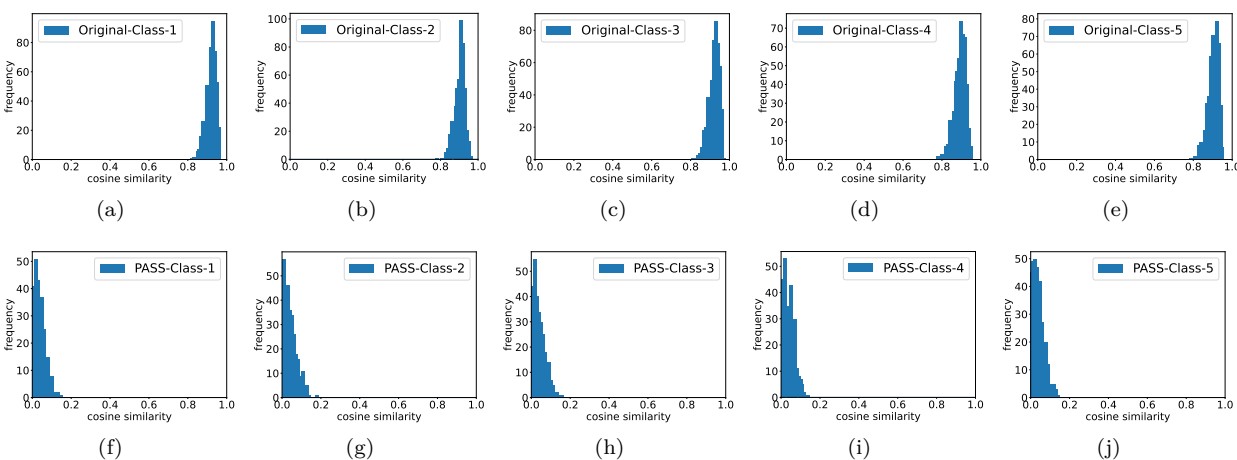

Figure 5: **Top row**: histograms of the original cosine similarity $\bar{a}_{k,i}$ (Eq. 2) between each class's prototype and the real samples (top). The augmented samples of YoooP+ are drawn from the original histograms. **Bottom row:** histograms of the cosine similarity between each class's prototype and the augmented samples for PASS. PASS fails to preserve the distribution of the original data.

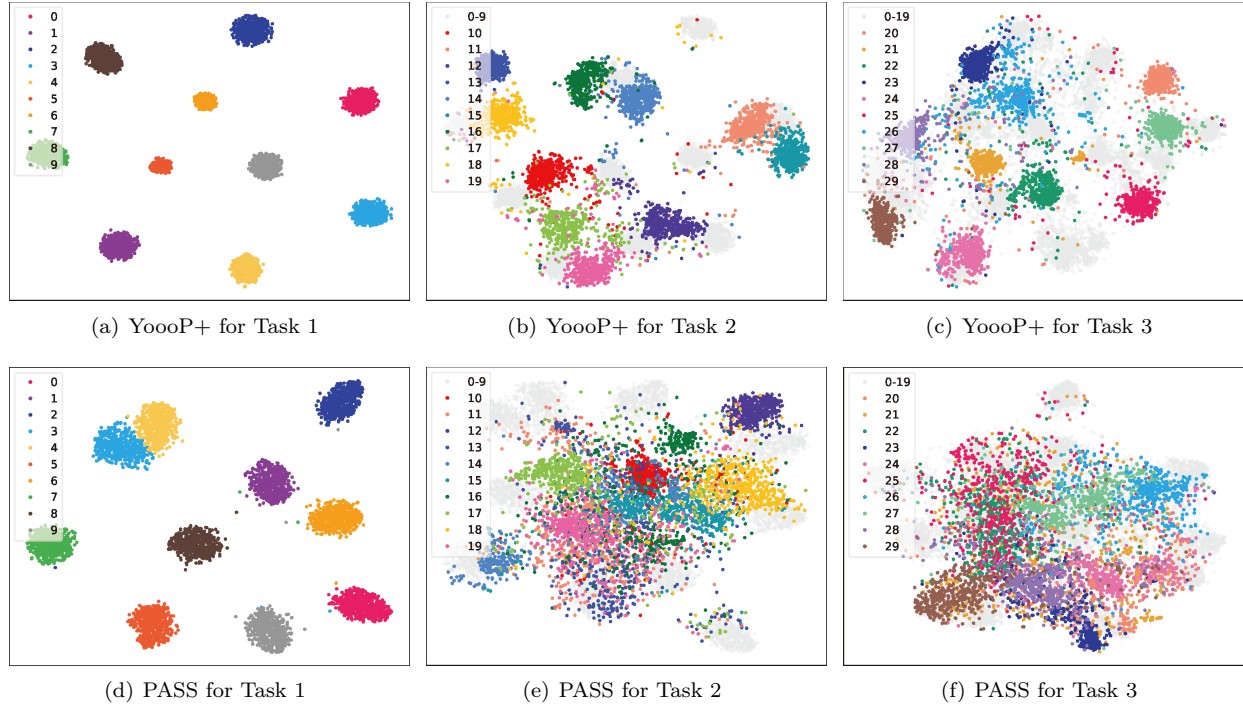

Figure 6: Visualization of the distribution of representations encoded by YoooP+ and PASS on CIFAR-100 base-0 phase 10 setting. The lighter gray points in "Task 2" and "Task 3" represent the distribution of the previous tasks' data.

## 4.4 Ablation Studies

We conduct ablation studies to assess the influence of three key components on model performance: prototype optimization (P), synthetic data replay, and model interpolation (MI). Fig. 7 presents the results on CIFAR-100 under the zero-base setting with 10 phases. In particular, YoooP (Class Mean) uses the class-mean for prototype generation (similar to PASS) while still employing

the prototype optimization loss in Eq. 3, YoooP (-P) omits prototype optimization, YoooP (-MI) excludes model interpolation, and YoooP (-P-MI) removes both. From Fig. 7, prototype optimization proves crucial, as YoooP (-P) suffers a substantial drop in accuracy compared to the full YoooP. When comparing YoooP and YoooP (Class Mean), YoooP achieves slightly higher accuracy while utilizing a mini-batch attentional mean-shift-based method that requires fewer stored sample embeddings than the conventional class-mean approach (see Appendix F). Moreover, YoooP+ (which incorporates prototype augmentation) further improves prediction accuracy relative to YoooP. Although model interpolation (MI) contributes a modest performance boost by retaining prior knowledge and ensuring current-task performance, the difference between YoooP (-P) and the other variants indicates that prototype optimization remains the most critical factor.

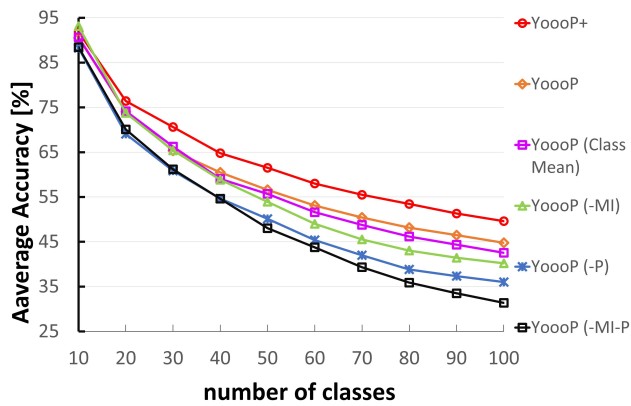

Figure 7: Ablation study of different components in YoooP+. "-M" means without model interpolation, "-P" means without prototype optimization.

## 5   Conclusion

In this work, we developed two non-exemplar-based methods, YoooP and YoooP+, for class-incremental learning. Specifically, YoooP only needs to store and replay one optimized prototype for each class without generating synthetic data from stored prototypes. As an extension of YoooP, YoooP+ proposed to create synthetic data from the stored prototypes and the stored distribution of cosine similarity with the help of a high-dimensional rotation matrix. The evaluation results on multiple benchmarks demonstrated that both YoooP and YoooP+ can significantly outperform the baselines in terms of accuracy and average forgetting. Importantly, this work offered a new perspective on optimizing class prototypes for exemplar-free CIL. We also show more experimental results in Appendix G.

## Acknowledgments

Research reported in this paper was sponsored in part by NSF CPS 2311086, NSF CIRC 716152, and Faculty Research Grant at William & Mary 141446.

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

# A  YoooP and YoooP+ Algorithms

We summarize the proposed YoooP and YoooP+ algorithms as follows. Regarding parameters $S$ steps and $R$ iterations in Algorithm 1, they are determined by the ratio of Dataset Size to Mini-batch size. More specifically, $S$ is the number of total steps per epoch for each task. For the number of steps $R$ used to update the prototypes in each task, we simply set $R = S$ in our experiments. For computing prototype $p_k$, we randomly choose one sample from each class $k$ and use its embedding as the initial prototype. Our prototype update method is an extension of the classical mean-shift algorithm Cheng (1995) that replaces the pre-defined kernel similarity with a learnable attention score. As stated in Cheng (1995), random initialization of $p_k$ can guarantee the independence of the solution to the initialization problem.

---

**Algorithm 1:** YoooP and YoooP+

---

**input**      : Training data $\mathcal{D}_{1:T}$ with classes $C_{1:T}$, epochs $E$, steps $S$, iterates $R$, number of bins $N_b$ learning rate
        $\eta$, $\alpha$, $\beta$

**initialize** : Memory $\mathcal{M} \leftarrow \emptyset$, Distribution $\mathcal{P} \leftarrow \emptyset$, $\theta, w$

1 **for** $t = 1 \rightarrow T$ **do**
2  **for** $epoch=1 \rightarrow E$ **do**
3    Compute features $z_i = F(x_i; \theta)$ for $(x_i, y_i) \in \mathcal{D}_t$;
4    Compute prototype $p_k$ for every class $k \in C_t$ by iterating Eq. 1 for $R$ iterations;
5    Save prototypes: $\mathcal{M} \leftarrow \mathcal{M} \cup \{(p_k, k) : k \in C_t\}$;
6    Save distribution of cosine similarity using a histogram: $\mathcal{P} \leftarrow \mathcal{P} \cup \{P(\bar{a}_{k,i}) : k \in C_t\}$ ;
7    **for** $step=1 \rightarrow S$ **do**
8      Draw a mini-batch of data $(x_i, y_i) \sim \mathcal{D}_t$;
9      **if** *YoooP+* **then**
10        Data Synthesis: create a mini-batch of data $(z'_i, k)$ of previous classes from prototypes
          $\{p_k : k \in C_{1:t-1}\}$ and the distribution of cosine similarity $\{P(\bar{a}_{k,i}) : k \in C_t\}$ in Eqs. 7,8;
11        Compute loss $L_t(\theta, w)$ in Eq. 11 on the two mini-batches;
12      **else**
13        Draw a mini-batch of prototypes $(p_k, k)$ from $k \in C_{1:t-1}$;
14        Compute loss $L_t(\theta, w)$ in Eq. 6 on the two mini-batches;
15      Update feature extractor: $\theta \leftarrow \theta - \eta \nabla_\theta L_t(\theta, w)$;
16      Update classifier for $k \in C_t$: $w_k \leftarrow w_k - \eta \nabla_{w_k} L_t(\theta, w)$;
17      Update classifier for $k \in C_{1:t-1}$: $w_k \leftarrow w_k - \eta \left(\alpha \nabla_{w_k} L_t(\theta, w)\right)$;
18   Model interpolation: $\theta \leftarrow (1-\beta)\theta' + \beta\theta$;
19   Save current task model as $\theta' \leftarrow \theta$;

**output**   : Feature extractor $F(\cdot; \theta)$ and classifier $G(\cdot; w)$

---

# B  Effect of Hyper-parameters

We also explore some important hyper-parameters on the estimation accuracy of YoooP+ as follows.

### B.1  Effect of $\lambda$ in mean-shift method.

Tab. 4 shows the impact of hyper-parameter $\lambda$ from Eq. 1 in mean-shift-based method on CIFAR100. We can observe that as $\lambda$ increases from 0.3 to 0.9, the prediction accuracy of YoooP+ almost remains the same. Thus, we can conclude that the proposed method is not sensitive to $\lambda$ that controls the steps in the attentional mean-shift-based algorithm. We choose $\lambda = 0.6$ in our experiments.

### B.2  Effect of parameter $\beta$ in model interpolation.

We also investigate the influence of hyperparameter $\beta$ from Eq. 12 in model interpolation on the prediction accuracy, as shown in Fig. 8 (a). It can be seen that when $\beta = 0.6$, the proposed method has the best performance. We change the $\beta$ from 0.3 to 0.8, the performance just slightly drops when the $\beta = 0.8$. Thus we conclude that the proposed method is not sensitive to $\beta$.

Table 4: Effect of $\lambda$ on the accuracy of different tasks on CIFAR-100 with base-0 10 phases setting. We can see that our method is not sensitive to $\lambda$.

| $\lambda$ | Task 1 | Taks 2 | Task 3 | Task 4 | Task 5 | Task 6 | Task 7 | Task 8 | Task 9 | Task 10 |
|---|---|---|---|---|---|---|---|---|---|---|
| Base-0-10 phases Average Accuracy on CIFAR-100 with different $\lambda$. [%] ↑ | | | | | | | | | | |
| 0.3 | 92.99 | 76.67 | 70.79 | 63.91 | 60.36 | 57.57 | 55.69 | 53.80 | 51.26 | 49.39 |
| 0.4 | 91.85 | 76.45 | 70.65 | 64.57 | 61.34 | 57.84 | 55.26 | 53.25 | 51.15 | 49.59 |
| 0.5 | 91.09 | 76.12 | 70.02 | 63.97 | 61.85 | 58.08 | 55.79 | 53.83 | 51.53 | 49.15 |
| 0.6 | 91.83 | 76.42 | 70.62 | 64.78 | 61.51 | 57.99 | 55.49 | 53.44 | 51.31 | 49.60 |
| 0.7 | 91.77 | 76.67 | 71.03 | 64.83 | 61.45 | 57.95 | 55.54 | 53.48 | 51.16 | 49.54 |
| 0.8 | 91.89 | 76.71 | 70.77 | 64.86 | 61.55 | 58.07 | 55.60 | 53.45 | 51.31 | 49.63 |
| 0.9 | 91.84 | 76.74 | 70.74 | 64.86 | 61.59 | 58.14 | 55.51 | 53.46 | 51.13 | 49.56 |

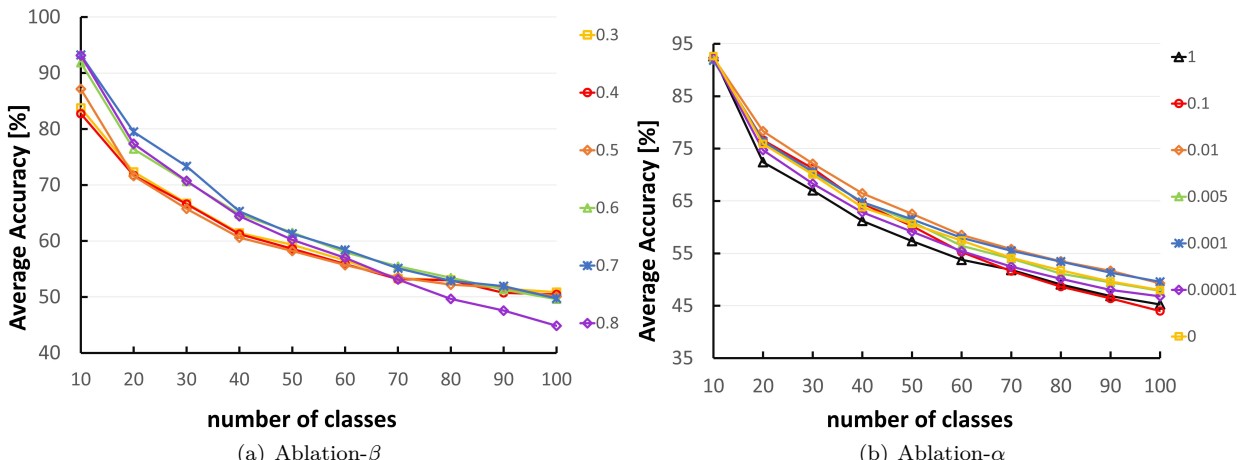

(a) Ablation-$\beta$          (b) Ablation-$\alpha$

Figure 8: The ablation study of $\alpha$ and $\beta$ on the prediction results.

## B.3 Effect of small factor $\alpha$.

Lastly, we study the effect of small factor $\alpha$ from Eq. 13 in the "Partial Freezing" of the classifier on the model performance. As illustrated in Fig. 8 (b), it can be observed that when $\alpha$ is 0 or larger than 0.005, the prediction accuracy will drop slightly. As $\alpha$ is a very small value, such as 0.001, the proposed method has the best performance. Hence, we choose $\alpha = 0.001$ in our experiment.

## C Evaluation on Sub-ImageNet Dataset

In this section, we further evaluate our methods on a larger Sub-ImageNet dataset created by randomly selecting 100 classes Deng et al. (2009). This experiment demonstrates how YoooP and YoooP+ compare against various non-exemplar-based approaches under different settings. We exclude IL2A from the comparison due to its prohibitively high GPU memory requirements on Sub-ImageNet.

Table 5: Average incremental accuracy (AIA) and average accuracy after training the last task (AA) of the proposed YoooP/YoooP+ and baselines on Sub-ImageNet under zero-base setting and half-base setting with 10 phases."b0-10" means zero-base with 10 phases, "half-10" means half-base with 10 phases. (⋆) denotes methods using "Nearest Class Mean"(NCM) classifier. **Bold**: the best among non-exemplar methods. Underline: the second best among non-exemplar methods.

| Sub-ImageNet | AIA [%]↑ | | AA [%]↑ | |
|---|---|---|---|---|
| Methods | b0-10 | half-10 | b0-10 | half-10 |
| LwF Li & Hoiem (2017) | 30.64 | 27.46 | 15.62 | 16.62 |
| SSRE Zhu et al. (2022) | 41.12 | 60.41 | 24.43 | 51.20 |
| FeTrIL Petit et al. (2023) | 46.73 | 72.55 | 28.84 | 64.06 |
| ⋆ADC Goswami et al. (2024) | 65.90 | 53.18 | 50.66 | 40.26 |
| PASS Zhu et al. (2021b) | 44.10 | 64.57 | 27.70 | 53.54 |
| YoooP (Ours) | 68.28 | 75.86 | 52.86 | 68.96 |
| YoooP+ (Ours) | **70.35** | **78.24** | **56.20** | **70.54** |

As shown in Tab. 5 and Fig. 9, YoooP and YoooP+ consistently outperform the non-exemplar baselines in both zero-base and half-base configurations. In line with our earlier findings (Sec. 4.1), FeTrIL continues to perform well in the half-base setting but fails in the zero-base scenario, likely because it cannot learn sufficient

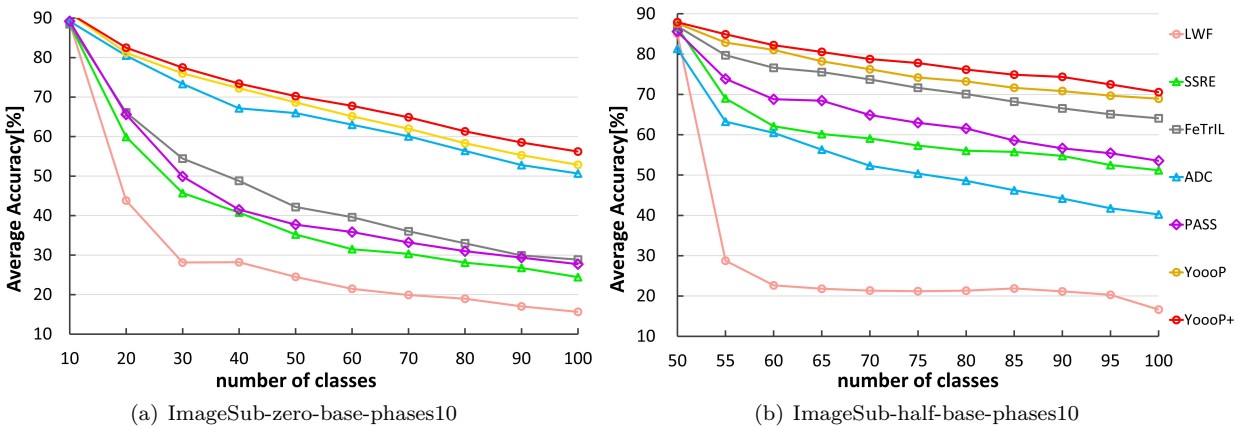

(a) ImageSub-zero-base-phases10      (b) ImageSub-half-base-phases10

Figure 9: Accuracy comparison of different non-exemplar-based and exemplar-based methods on ImageNet-Subset under different settings using 3 random seeds.

initial knowledge when starting with no base classes. Meanwhile, ADC with an NCM classifier excels under zero-base conditions but suffers a large performance drop in the half-base setting. This is primarily due to the substantial distribution shift between tasks in the half-base scenario, which is especially pronounced on a large-scale dataset like Sub-ImageNet.

## D Evaluation on 20 phases setting.

We further evaluate our proposed methods on Tiny-ImageNet under more extended incremental settings, specifically 20-phase scenarios in both zero-base and half-base configurations. As shown in Tab. 6, FeTrIL, FeCAM, and DS-AL, continue to perform well in the half-base setting (earning second place) but experience a severe performance drop in the zero-base setting. Meanwhile, ADC with an NCM classifier achieves the second-best AIA under the zero-base setting but is consistently outperformed by YoooP and YoooP+, particularly in the half-base configuration. Comparing to the PASS, the performance of our proposed methods surpass it in a clear gap across various settings. These findings reinforce our earlier observations that relying on the performance of the initial tasks (as in FeTrIL, FeCAM, EFC, and DS-AL) or an NCM classifier (as in ADC) limits adaptability when the task distribution shifts substantially. By contrast, YoooP and YoooP+ maintain strong performance across both zero-base and half-base 20-phase settings, highlighting their robustness in more demanding incremental learning scenarios.

Table 6: Average incremental accuracy (AIA) and average accuracy after training the last task (AA) of the proposed YoooP/YoooP+ and baselines on TinyImageNet under zero-base setting and half-base setting with 20 phases. "b0-20" means zero-base with 20 phases, "half-20" means half-base with 20 phases. (⋆) denotes methods using "Nearest Class Mean"(NCM) classifier. **Bold**: the best among non-exemplar methods. Underline: the second best among non-exemplar methods.

| TinyImageNet | AIA [%]↑ | | AA [%]↑ | |
|---|---|---|---|---|
| Methods | b0-20 | half-20 | b0-20 | half-20 |
| LwF Li & Hoiem (2017) | 22.93 | 13.39 | 9.57 | 5.62 |
| SSRE Zhu et al. (2022) | 25.78 | 46.20 | 15.25 | 38.84 |
| IL2A Zhu et al. (2021a) | 25.39 | 39.90 | 17.73 | 29.43 |
| FeTrIL Petit et al. (2023) | 31.84 | 48.96 | 18.38 | 41.44 |
| DS-AL Zhuang et al. (2024) | 33.74 | 58.17 | 24.58 | 50.79 |
| EFC Magistri et al. (2024) | 42.31 | 56.01 | 28.27 | 47.21 |
| AdaGauss Rypeść et al. (2024) | 44.21 | 50.31 | 30.26 | 41.95 |
| ⋆FeCAM Goswami et al. (2023) | 38.61 | **59.32** | 25.30 | **54.61** |
| ⋆ADC Goswami et al. (2024) | 46.75 | 34.64 | 29.24 | 25.70 |
| PASS Zhu et al. (2021b) | 33.76 | 33.24 | 19.36 | 25.21 |
| YoooP (Ours) | 46.00 | 48.19 | 30.97 | 39.31 |
| YoooP+ (Ours) | **48.36** | 51.27 | **31.26** | 42.28 |

## E Compare with Exemplar-based methods

To showcase the strong performance of our proposed methods, we also compare them with several exemplar-based approaches, including iCaRL Rebuffi et al. (2017), BiC Wu et al. (2019), and WA Zhao et al. (2020), under zero-base and half-base settings with 5 and 10 phases on TinyImageNet. Following prior works Hou et al. (2019); Rebuffi et al. (2017), these exemplar-based methods store 20 raw samples per old class, selected

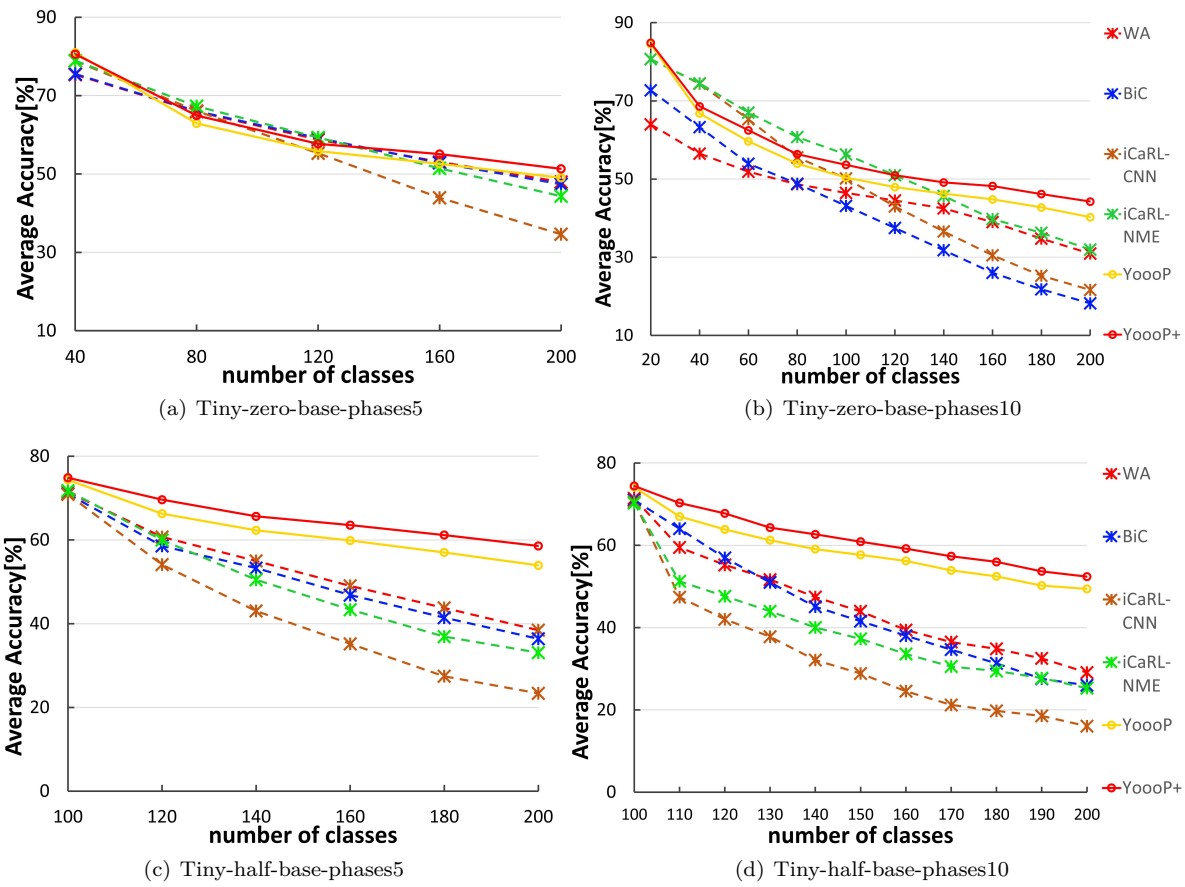

Figure 10: Accuracy comparison of different exemplar-based methods on TinyImageNet under different settings using 3 random seeds.

via herd election Rebuffi et al. (2017). It is important to note that exemplar-based methods typically achieve higher accuracy than non-exemplar-based methods, as they can store raw data and thus preserve richer information Rebuffi et al. (2017); Hou et al. (2019). This makes a fair comparison particularly challenging.

Fig. 10 shows the performance of different approaches on TinyImageNet. Notably, although our YoooP and YoooP+ do not store raw data, they still achieve comparable performance in zero-base settings and even surpass some exemplar-based methods in half-base settings. This result underscores the effectiveness of our prototype optimization and augmentation strategies, demonstrating that even without raw data storage, our approach can overcome the inherent advantages of exemplar-based methods under certain conditions.

# F   Evaluation of Additional Storage Requirements

We further evaluate the additional storage requirements of our proposed mini-batch attentional mean-shift-based method presented in Eq. 1. Specifically, this metric quantifies the number of sample embeddings that must be stored for generating prototypes of classes. In our approach, only the embeddings from a mini-batch are retained, whereas the conventional class-mean method requires storing the embeddings of all samples within each class. Tab. 7 presents a comparison of these storage requirements for individual tasks under different settings on TinyImageNet dataset. As shown in Tab. 7, our methods require significantly fewer stored embeddings than class-mean approaches, demonstrating a clear advantage in terms of memory efficiency.

Table 7: Number of stored embeddings utilized for generating prototype in each task of our proposed method and class mean method in the baselines under different settings on TinyImageNet.

| Additional Storage Requirements under different settings on TinyImageNet: # ↓ | | | |
|---|---|---|---|
| Settings | Our method | | Class-mean |
| | batch size: 128 | batch size: 256 | |
| base0-phases5 | **128** | 256 | 20000 |
| base0-phases10 | **128** | 256 | 10000 |
| half-base-phases5 | **128** | 256 | 10000 |
| half-base-phases10 | **128** | 256 | 5000 |

## G   Evaluation on the confusion matrix

To effectively demonstrate the results of Class-Incremental Learning (CIL), we conduct a comparison of the classification confusion matrices for PASS, YoooP, and YoooP+ on CIFAR-100 under the zero-base 10-phase setting. The confusion matrix, as shown in Fig. 11, illustrates correct predictions on the diagonal and misclassifications in off-diagonal entries.

Comparing Fig. 11(a) and Fig. 11(b), we observe that the diagonal entries' temperatures in YoooP are higher than those in PASS, which means the accuracy is higher, particularly for classes 20 to 100. This indicates that YoooP excels in learning the current task while minimizing forgetting of the old tasks. Consequently, YoooP exhibits enhanced plasticity and stability compared to PASS.

Further comparing Fig. 11(b) and Fig. 11(c), we notice that the diagonal entries' temperatures for classes 0 to 40 in YoooP+ are higher than those in YoooP. Notably, YoooP tends to classify more into recent tasks. YoooP+ effectively mitigates this tendency and reduces bias through the proposed prototype augmentation, resulting in superior performance.

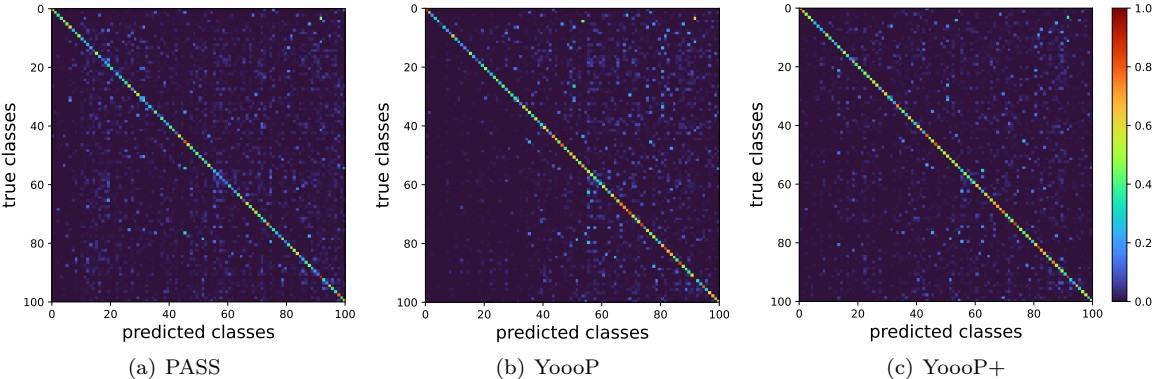

(a) PASS          (b) YoooP          (c) YoooP+

Figure 11: Confusion matrix comparison of different methods under CIFAR-100 base0-10 setting.

# H    Evaluation on the efficiency

Table 8: Efficiency on TinyImageNet under zero-base 10 phases setting, One GPU NVIDIA RTX A6000, BS=256, M=$10^6$, G=$10^9$, T=$10^{12}$.

| Methods | MACs↓ | Parameters↓ | GPU-Memo↓ | Time/Epoch↓ |
|---------|-------|-------------|-----------|-------------|
| YoooP+ | 571.26 G | 11.17 M | 9.86 GB | 22.07 s |
| YoooP | 571.26 G | 11.17 M | 9.53 GB | 17.10 s |
| ADC | 571.26 G | 11.17M | 9.53 GB | 17.00 s |
| PASS | 2.28 T | 11.18 M | >50 GB | 40.64 s |
| FeTril | 571.26 G | 11.17 M | 10.72 GB | 349.15 s |
| SSRE | 633.31 G | 12.40 M | 13.23 GB | 50.17 s |
| LwF | 571.26 G | 11.17 M | 9.53 GB | 16.90 s |
| IL2A | 2.74 T | 11.17 M | >50 GB | 97.4 s |

To show that both YoooP and YoooP+ are designed to be lightweight, we provide a detailed analysis of the computational cost and implementation details for prototype optimization and synthetic data replay in Tab. 8. As shown in the table, our YoooP+ method achieves competitive or even lower MACs, GPU memory, and training time per epoch compared to other baselines (e.g., ADC, PASS, FeTrIL). Specifically, YoooP+ requires 571.26 G MACs, 9.86 GB GPU memory, and 22.07 seconds per epoch, demonstrating that prototype optimization and synthetic data replay introduce minimal overhead.

