# OpenReview forum: "YoooP: You Only Optimize One Prototype per Class for Non-Exemplar Incremental Learning"
_TMLR — Accepted by TMLR_

### Review · Reviewer_obox · 2025-04-29

**Summary Of Contributions:**

This paper tackles the non-exemplar class-incremental learning problem, an important problem in the machine learning field. The authors present an enhanced prototypical network framework with novel loss functions specifically designed for this problem. The proposed approach demonstrates competitive performance through evaluations on benchmark datasets, outperforming existing methods.

**Audience:**

Yes

**Claims And Evidence:**

Yes

**Requested Changes:**

See Weaknesses.

**Strengths And Weaknesses:**

Pros:
1. This paper tackles the non-exemplar class-incremental learning problem, an important problem in the machine learning field.
2. The proposed method is rigorously evaluated on standard benchmark datasets and demonstrates better performance compared to existing approaches.
3. The paper proposed a novel prototype augmentation technique, which is effective.

Cons:

1.	Comparisons with other SOTA methods are missing, such as [1] and [2]. For example, on Cifar100 b50in10 setting, the AIA of DS-AL[1] is 68.4%, but the proposed method is 67.4%.
2.	Multiple things are combined in the Yooop+ method - three loss components and two tricks, where feature distillation is quite a strong regularization, preventing forgetting. However, mixing feature distillation with model interpolation is confusing.
3.	The ablation is not clear about every single participant. I want to know the contribution of the feature distillation loss item and "Partial Freezing" of the Classifier.
4.	The method's reliance on synthetic data increases computation time, yet the paper omits runtime comparisons with existing methods.
5.	The method proposes a plug-in module so it can be combined with other methods. I hope to see the results.
6.	While the approach leverages prototypes for loss construction, both the backbone and the classifier remain trainable throughout the learning process. The current title may not fully capture the scope of the proposed method.

[1] DS-AL: A Dual-Stream Analytic Learning for Exemplar-Free Class-Incremental Learning. AAAI2024

[2] Task-recency bias strikes back: Adapting covariances in Exemplar-Free Class Incremental Learning. NeurIPS2024

---

> ### Author Response · Authors · 2025-06-10
>
> Q1: Comparisons with other SOTA methods are missing, such as [1] and [2]. For example, on Cifar100 b50in10 setting, the AIA of DS-AL[1] is 68.4%, but the proposed method is 67.4%.
> -----------------
> **A1:** Per your suggestion, we implemented the baseline as per the original paper and code, and compared our methods with [1], [2] on CIFAR-100 and TinyImageNet with 10 phases under zero-base and half-base settings. The results are presented in Tab. 1.
> Table 1: Evaluation on CIFAR-100 and TinyImageNet.
> |  Methods  | CIAFR b0-10 AIA | CIAFR b-10 AA | CIAFR half-10 AIA | CIAFR half-10 AA | Tiny b0-10 AIA | Tiny b0-10 AA | Tiny half-10 AIA | Tiny half-10 AA |
> |-|-|-|-|-|-|-|-|-|
> | YoooP+ | 63.30\% | 49.60\% | 61.83\% | 50.81\% | 56.47\% | 44.25\% | 61.72\% | 52.40\% |
> | DS-AL | 56.70\% | 42.76\% | 68.37\% | 61.23\% | 47.24\% | 33.71\% | 48.05\% | 42.54\% |
> | AdaGauss | 59.13\% | 46.32\% | 63.50\% | 54.82\% | 50.51\% | 36.52\% | 56.00\% | 47.90\% |
>
> We observe that YoooP+ consistently outperforms DS-AL and AdaGauss under zero-base settings on both datasets and half-base settings on TinyImageNet, where the distribution shift is significant or the initial knowledge is limited, while slightly lower on CIFAR-100 under the half-10 setting.
>
> Regarding DS-AL and AdaGauss, in Sec. 4.3 we analyzed methods like FeTrIL, which share a similar setting with DS-AL by relying on the feature extractor learned from the first task. These methods perform well under half-base settings on CIFAR-100, where the first task already includes half of all classes, reducing distribution shift and stabilizing the extractor. However, they struggle under zero-base settings due to larger distribution shifts and lack of prior knowledge. AdaGauss models each class with a Gaussian distribution for generating synthetic data, but faces a trade-off: forming tight clusters benefits classification while maintaining feature diversity is necessary to avoid covariance collapse. This trade-off can compromise either sampling stability or decision boundary clarity, especially with significant distribution shifts. Consequently, AdaGauss performs well only on CIFAR-100 with half-base setting but struggles with zero-base setting and both settings on TinyImageNet. In contrast, our method explicitly optimizes each class prototype and models intra-class distributions with an angular histogram, naturally balancing cluster compactness and diversity. This design enables YoooP+ to maintain stable replay and clear class boundaries without suffering covariance collapse.
>
> In terms of buffer efficiency, our method significantly reduces storage overhead compared to DS-AL and AdaGauss. DS-AL requires storing a large embedding buffer containing all sample features from the base task (shape: [N, D], e.g., N=8192 in CIFAR-100), while AdaGauss stores both class means and covariance matrices for **each class** (shape: [D] + [D, D]). In contrast, YoooP+ only stores a single prototype (shape: [D]) and a 1 dim angular histogram **per class** (shape: [$N_b$], e.g., $N_b=100$ in our paper), resulting in substantially lower memory consumption and better scalability.
>
> In summary, YoooP+ demonstrates clear advantages in accuracy, memory efficiency, and generalization, particularly in challenging scenarios with significant distribution shifts.
>
> [1] DS-AL (Zhuang et al., AAAI 2024)
> [2] AdaGauss (Rypeść et al., NeurIPS 2024)

---

> ### Author Response · Authors · 2025-06-10
>
> Q2: Mixing feature distillation with model interpolation is confusing.
> ------------------
> **A2:** We clarify that feature (knowledge) distillation (KD) and model interpolation (MI) serve complementary purposes in our exemplar-free setting. KD (Eq.12) constrains the feature extractor on current-task samples to preserve representation consistency with the previous model. However, since KD operates only on current-task data, it cannot fully prevent parameter drift due to the absence of old-task examples. To address this, MI (Sec.4.3) acts at the parameter level to globally stabilize $\theta$ by softly blending it with $\theta_{t-1}$, further reducing forgetting. Their combination is theoretically sound and practically effective, as confirmed by our ablations (Fig.9). We will clarify this distinction in the final version.
>
> Q3: The ablation is not clear enough.
> ------------------
> **A3:** Per your suggestion, while the feature distillation (KD) loss is a widely adopted regularization in non-exemplar CIL methods (e.g., [1,2,3]) and the partial freezing of the classifier is a standard stabilizing improvement (e.g., [1]), we added the ablation study based on YoooP. As shown in Tab.2, the perofrmance of YoooP (w/o KD) drops a lot (26.69\% on AIA and 31.27\% on AA) since the KD plays a basic and important role for incremenatlly updating the backbone in CIL, while the YoooP (w/o PF) drops a little (1.89\% on AIA and 1.3\% on AA), but it still can improve the performance.
>
> Table 2: Evaluation on CIFAR-100 with zero-base 10 phases.
> |  CIFAR-100  | AIA | AA |
> |  ----  | ----  | ---- |
> | YoooP | 58.99\% | 44.78\% |
> | YoooP (w/o KD) | 32.30\% | 13.51\% |
> | YoooP (w/o PF) | 57.10\% | 43.48\% |
>
> [1] LwF (Li et al., ECCV 2016)
> [2] LUCIR (Hou et al., CVPR 2019)
> [3] PASS (Zhu et al., CVPR 2022)
>
> Q4: Omits runtime comparisons with existing methods.
> ------------------
> **A4:** Per your suggestion, we compare the efficiency for YoooP/YoooP+ in Tab.3. Our methods are more efficient compared to the baselines.
>
> Table 3: Efficiency on TinyImageNet under zero-base 10 phases setting, One GPU NVIDIA RTX A6000, BS=256, M=$10^6$, G=$10^9$, T=$10^{12}$.
> | Method | MACs↓ | Parameters↓ | GPU-Memo↓ | Time/Epoch↓ |
> |----------|-----------|-------------|-----------|-------------|
> | YoooP+ | 571.26 G | 11.17 M | 9.86 GB | 22.07 s |
> | YoooP | 571.26 G | 11.17 M | 9.53 GB | 17.10 s |
> | ADC | 571.26 G | 11.17M | 9.53 GB | 17.00 s |
> | PASS | 2.28 T | 11.18 M | >50 GB | 40.64 s |
> | FeTril | 571.26 G | 11.17 M | 10.72 GB | 349.15 s |
> | SSRE | 633.31 G | 11.17 M | 10.23 GB | 63.58 s |
> | LwF | 571.26 G | 11.17 M | 9.53 GB | 16.90 s |
> | IL2A | 2.74 T | 11.17 M | >50 GB | 97.4 s |
>
> Q5: The results of the proposed method combined with other methods.
> ------------------
> **A5:** We would like to clarify that we did not claim our method to be a plug-in module in the paper; rather, YoooP/YoooP+ is presented as an integrated approach tailored for non-exemplar CIL. While theoretically compatible with parameter-based regularization like EWC, such integration requires careful tuning. We explored EWC in Tab. 4 as a representative example, but extensive exploration of other combinations is beyond the scope of this work. We plan to investigate this further in future work.
>
> Table 4: Evaluation on CIFAR-100 with zero-base 10 phases.
> |  CIFAR-100  | AIA | AA |
> |  ----  | ----  | ---- |
> | EWC | 48.02\% | 30.13\% |
> | EWC+YoooP | 59.01\% | 45.03\% |
> | EWC+YoooP+ | 63.58\% | 49.86\% |
>
> As shown in the Tab. 4, our YoooP/YoooP+ improves the EWC a lot, but the EWC impact the performance of YoooP/YoooP+ a little, which highlight that our YoooP/YoooP+ is presented as an integrated approach tailored for non-exemplar CIL.
>
> Q6: The current title may not fully capture the scope of the proposed method.
> -----------------
> **A6:** Thanks for your suggestion. We will consider refining the title to better capture the full scope of the method in the final version.

---

### Review · Reviewer_7TYA · 2025-05-24

**Summary Of Contributions:**

The paper focus on non-exemplar continual learning using class-wise prototypes and discuss the limitations of methods like PASS which uses a single class centroid as the prototype and replays synthetic features. The authors propose a mini-batch attentional mean shift-based method to construct more representative prototypes. The authors also propose to generate synthetic features by exploiting the angular distribution between each class’s prototypes and its features. The proposed method outperforms competitive non-exemplar IL methods.

**Audience:**

Yes

**Broader Impact Concerns:**

The paper does not have ethical concerns.

**Claims And Evidence:**

Yes

**Requested Changes:**

I would request the authors to address the points in weaknesses mentioned above.

**Strengths And Weaknesses:**

Strengths -
1. The paper proposes an interesting approach for optimizing prototypes and generating synthetic samples around prototypes to obtain distributions closer to original distributions.

Weaknesses-
1. The motivation is heavily based on a single paper - PASS [CVPR 2021] since FeTrIL does not augment prototypes and instead samples from a gaussian distribution using class distributions with the assumption that similar classes have similar feature distributions. The paper is not discussed or based on more relevant and recent works on non-exemplar IL [a,b,c].

2. FeCAM [a] proposed to use class distributions in the form of class covariances to better represent classes and then used a Mahalanobis-distance based classifier for non-exemplar IL. The authors ignore any discussion with covariance based methods [a,b] and does not include them in the experiements. [a, b, c] performs better than FeTrIL in most settings and are relevant baselines for non-exemplar IL. For instance, it is not clear why one needs to optimize the prototypes or generate synthetic samples when they can simply use the class covariances and adapt them continually?

3. While prototype optimization is interesting, it is not clear why the authors do not consider using class covariances in any form to improve class representations. The motivation for the proposed class centroid optimization is not so coherent in the paper. Why is the average of the class features not a good prototype? Can the authors show some analysis on how distant is the optimized prototype from the class feature centroid?

4. How good is the proposed generation of synthetic samples compared to sampling features from a gaussian using the class mean and covariances?

5. The method has several components and the implementation in Algorithm 1 says that it involves more complexity. The training needs several steps in each epoch which is computation intensive. Can the authors discuss more on this? The improvements does not look very impressive given the complexity and combination of several components in the method.

6. How is the inference done? While it is a prototype-based method and the authors stress on using one optimized prototype per class, it is not using the prototypes for inference. How good is the performance if you use the optimized prototypes for NCM classification? Is this better than standard NCM with class feature centroids?

[a] "Fecam: Exploiting the heterogeneity of class distributions in exemplar-free continual learning." Advances in Neural Information Processing Systems 36 (2023).

[b] "Task-recency bias strikes back: Adapting covariances in exemplar-free class incremental learning." Advances in Neural Information Processing Systems 37 (2024).

[c] "Elastic Feature Consolidation For Cold Start Exemplar-Free Incremental Learning." The Twelfth International Conference on Learning Representations, 2024.

---

> ### Author Response · Authors · 2025-06-10
>
> Q1: The paper is not discussed or based on more relevant and recent works on non-exemplar IL [1,2,3].
> ------------------
> **A1:** Per your suggestion, we implemented the baseline as per the original paper and code, and compared our methods with [1], [2], [3] on CIFAR-100 and TinyImageNet with 10 phases under zero-base and half-base settings. The results are presented in Tab. 1.
>
> Table 1: Evaluation on CIFAR-100 and TinyImageNet.
> |  Methods  | CIAFR b0-10 AIA | CIAFR b-10 AA | CIAFR half-10 AIA | CIAFR half-10 AA | Tiny b0-10 AIA | Tiny b0-10 AA | Tiny half-10 AIA | Tiny half-10 AA |
> |-|-|-|-|-|-|-|-|-|
> | YoooP+ | 63.30\% | 49.60\% | 61.83\% | 50.81\% | 56.47\% | 44.25\% | 61.72\% | 52.40\% |
> | FeCAM | 48.93\% | 32.11\% | 66.97\% | 59.93\% | 46.03\% | 32.02\% | 60.02\% | 50.64\% |
> | EFC | 59.96\% | 45.24\% | 64.88\% | 54.87\% | 48.03\% | 34.12\% | 53.18\% | 40.35\% |
> | AdaGauss | 59.13\% | 46.32\% | 63.50\% | 54.82\% | 50.51\% | 36.52\% | 56.00\% | 47.90\% |
>
> As shown in Tab. 1, YoooP+ consistently outperforms FeCAM, AdaGauss, and EFC in challenging scenarios such as zero-base settings on both datasets and half-base settings on TinyImageNet. While slightly lower on CIFAR-100 under the half-10 setting, YoooP+ still achieves comparable results. This is likely because half-10 includes half of the classes in the initial task, reducing distribution shifts and favoring methods like EFC and FeCAM that rely on strong initial task representations.
>
> Regarding methodological differences, although we initially discuss limitations of PASS, the core challenges apply equally to FeCAM, AdaGauss, and EFC.
>
> First, these are all high-dimensional Gaussian-based methods that assume class feature distributions can be approximated by Gaussians. However, real class distributions may not be Gaussian. Even if they can be approximated, as discussed in Sec. 4.3 and Fig. 1 of our paper, sampling from high-dimensional Gaussians in Euclidean space suffers from the curse of dimensionality—samples cluster on a thin shell, missing the true high-density regions. This undermines realistic data generation and increases forgetting.
>
> Second, for methods that store the mean and covariance, their real clusters around the prototype may overlap, leading to blurry boundaries and increased forgetting. Without prototype optimization, as demonstrated in our t-SNE visualizations (Sec. 4.3), prototype clusters can overlap, causing decision boundaries to be unclear, and decrease the classifictaion performance. Furthermore, if they store the overlapped distribution, the augmented prototypes are unrealistic, and increase forgetting.
>
> In contrast, our approach explicitly performs prototype optimization to build tight, well-separated clusters that better represent classes and are robust to drift. To store and recover real distributions, we record the angular (cosine similarity) histogram between each prototype and its class embeddings, and sample from this histogram during replay. This avoids the curse of dimensionality and preserves the true data distribution.
>
> Thus, with prototype optimization and angular distribution modeling, our method achieves consistent and superior performance across various settings.
>
> [1] FeCAM (Zhang et al., NeurIPS 2023)
> [2] EFC (Magistri et al., ICLR 2024)
> [3] AdaGauss (Rypeść et al., NeurIPS 2024)
>
> Q2: It is not clear why one needs to optimize the prototypes or generate synthetic samples when they can simply use the class covariances and adapt them continually?
> ------------------
> **A2:** Following our discussion in A1, we need prototype optimization to learn a tight cluster for the prototype with the clear boundaries, increasing the robutness for the future task drift. The real clusters around the prototype of covariance-based methods may overlap, leading to blurry boundaries and increased forgetting.
>
> The reason why we need generate synthetic samples is that simply using the class covariances and adapt them continually (e.g., FeCAM[1]) has the limited plasticity due to the frozen backbone. These approaches fundamentally depend on having good initial decision boundaries learned by the feature extractor, which requires sufficient initial knowledge. As shown in Tab. 1 in A1, FeCAM [1] perform well on the half-base setting but degrade significantly under the zero-base setting due to larger distribution shifts and the inability to adapt.
>
> Therefore, to better handle challenging CIL scenarios with large distribution shifts or limited initial knowledge, our approach explicitly performs prototype optimization to build compact, well-separated clusters and uses angular-based prototype augmentation for realistic data replay, rather than relying solely on covariance-based methods.
>
> [1] FeCAM (Zhang et al., NeurIPS 2023)

---

> ### Author Response · Authors · 2025-06-10
>
> Q3-1: While prototype optimization is interesting, it is not clear why the authors do not consider using class covariances in any form to improve class representations.
> ------------------
> **A3-1:** As we discussed in A1, we do not use class covariances because relying on them for class representations risks reduced performance and increased forgetting.
>
> First, using class means and covariances assumes that the original feature distribution is well-approximated by a high-dimensional Gaussian. However, in practice, this assumption often fails; real feature distributions are rarely perfect Gaussians. Sampling from an incorrect distribution during replay can degrade performance and amplify forgetting.
>
> Second, even if the true distribution were Gaussian, directly sampling from a high-dimensional Gaussian distribution stored via class covariances suffers from the curse of dimensionality. As discussed in Sec. 4.3 and shown in Fig. 1 of our paper, such sampling concentrates points on a thin shell at a fixed radius, making it difficult to capture the true high-density region and reconstruct the original distribution effectively.
>
> In contrast, YoooP+ stores the real angular (cosine similarity) distribution of each class in a histogram and samples directly from this during replay. This histogram accurately preserves the class-specific feature distribution and avoids the curse of dimensionality since distances in the angular space remain discriminative even as the dimension grows.
>
>
>
> Q3-2: The motivation for the proposed class centroid optimization is not so coherent in the paper.
> ------------------
> **A3-2:** We would like to clarify the motivation behind our prototype optimization and why it is both coherent and necessary for building robust class representations that can support continual learning, especially in settings with large task drift or limited initial knowledge.
>
> Prototype optimization in our work does not refer to directly optimizing the prototype itself, but rather to optimizing the network’s representation space so that each class forms a tight, well-separated cluster around its prototype. Without prototype optimization, the real clusters around the prototype may overlap, leading to blurry decision boundaries and increased forgetting. As demonstrated in our t-SNE visualizations (Sec. 4.3), overlapping clusters degrade classification performance. Moreover, if such overlapping distributions are stored, sampling augmented prototypes from these distributions becomes unrealistic, further exacerbating forgetting.
>
> Thus, our prototype optimization is a principled design choice that provides more robust, tight, and discriminative class representations, effectively mitigating drift and forgetting in challenging continual learning scenarios.
>
> Q3-3: Why is the average of the class features not a good prototype? Can the authors show some analysis on how distant is the optimized prototype from the class feature centroid?
> ------------------
> **A3-3:** The reason why the class-mean prototype is not a good choice is that it stores all embeddings of each class to compute the class-mean prototype. Instead of storing all embeddings of each class to compute the class-mean prototype, we propose a mini-batch attentional mean-shift-based method to approximate the high-density region of each class using only a small number of embeddings per mini-batch. As detailed in Sec. 3.2 and validated in our ablation study (Sec. 4.3) and the appendix, both class-mean prototypes and our mini-batch attentional mean-shift prototypes benefit significantly from prototype optimization and achieve similar performance. Our approach reduces the storage requirement while maintaining prototype quality, highlighting the importance of prototype optimization as a general strategy that consistently improves model performance.
>
>
> Q4: How good is the proposed generation of synthetic samples compared to sampling features from a gaussian using the class mean and covariances?
> ------------------
> **A4:** We found that it is not feasible to make a direct quantitative comparison between our angular histogram-based sampling and Gaussian-based sampling using class means and covariances. After prototype optimization, the feature clusters become highly compact and well-separated, causing the covariance matrices to be nearly singular (with diagonal values sometimes less than $1e^{-5}$ ). This leads to numerical instability and NaN values during sampling, making Gaussian-based sampling unreliable in our setting. This observation further highlights the need for our proposed angular histogram-based sampling, which preserves the true data distribution without suffering from such numerical issues, as discussed in A1, A2, and A3 above.

---

> ### Author Response · Authors · 2025-06-10
>
> Q5: The method has several components and the implementation in Algorithm 1 says that it involves more complexity. The training needs several steps in each epoch which is computation intensive. Can the authors discuss more on this? The improvements does not look very impressive given the complexity and combination of several components in the method.
> ------------------
> **A5:** We would like to clarify that both YoooP and YoooP+ are designed to be lightweight, using a standard single-stage training pipeline where prototype optimization is simply an additional loss term integrated into the main training loop—there is no multi-phase or separate-stage training required.
>
> As shown in Tab. 2, YoooP+ achieves comparable or even lower MACs, GPU memory, and training time per epoch than other baselines (e.g., ADC, PASS, FeTrIL). Specifically, YoooP+ maintains 571.26 G MACs, 9.86 GB GPU memory, and 22.07 seconds per epoch—comparable to or better than many existing methods. This demonstrates that YoooP+ achieves competitive performance with minimal additional computational cost.
>
> Table 2: Efficiency on TinyImageNet under zero-base 10 phases setting, One GPU NVIDIA RTX A6000, BS=256, M=$10^6$, G=$10^9$, T=$10^{12}$.
> | Method | MACs↓ | Parameters↓ | GPU-Memo↓ | Time/Epoch↓ |
> |----------|-----------|-------------|-----------|-------------|
> | YoooP+ | 571.26 G | 11.17 M | 9.86 GB | 22.07 s |
> | YoooP | 571.26 G | 11.17 M | 9.53 GB | 17.10 s |
> | ADC | 571.26 G | 11.17M | 9.53 GB | 17.00 s |
> | PASS | 2.28 T | 11.18 M | >50 GB | 40.64 s |
> | FeTril | 571.26 G | 11.17 M | 10.72 GB | 349.15 s |
> | SSRE | 633.31 G | 11.17 M | 10.23 GB | 63.58 s |
> | LwF | 571.26 G | 11.17 M | 9.53 GB | 16.90 s |
> | IL2A | 2.74 T | 11.17 M | >50 GB | 97.4 s |
>
>
> Q6-1: How is the inference done? While it is a prototype-based method and the authors stress on using one optimized prototype per class, it is not using the prototypes for inference.
> ------------------
> **A6-1:** We would like to clarify that inference in YoooP/YoooP+ is conducted using a commonly used backbone (e.g., ResNet-18) with a standard learnable fully-connected (FC) classifier, rather than a Nearest Class Mean (NCM) classification. We chose this design because, as discussed in Sec. 4.3 of the main paper, NCM assumes class distributions are well-separated and nearly isotropic around their centroids, which can be limiting in scenarios with significant distribution shifts or overlapping classes. Moreover, using an FC classifier enables YoooP/YoooP+ to learn more discriminative decision boundaries that better accommodate non-Gaussian, anisotropic, or complex feature distributions, ensuring robust performance across tasks.
>
> Q6-2: How good is the performance if you use the optimized prototypes for NCM classification? Is this better than standard NCM with class feature centroids?
> ------------------
> **A6-2:** Per your suggestion, we conducted a thorough ablation study comparing FC, NCM with our mini-batch attentional mean-shift-based prototypes, and NCM with class-mean prototypes. The results are shown in Tab. 3.
>
> Table 3: Evaluation on CIFAR-100.
> |  Methods  | CIAFR b0-10 AIA | CIAFR b-10 AA | CIAFR half-10 AIA | CIAFR half-10 AA |
> |-|-|-|-|-|
> | YoooP (FC) | 58.99\% | 44.78\% | 58.99\% | 47.09\% |
> | YoooP (NCM) | 53.75\% | 40.96\% | 58.84\% | 46.93\% |
> | YoooP (class-mean NCM) | 51.96\% | 40.24\% | 57.88\% | 45.87\% |
>
> It is important to note that both our mini-batch attentional mean-shift-based prototypes and class-mean prototypes are learned in a representation space shaped by prototype optimization and feature extractor training, comparing with a “standard NCM with class feature centroids” using non-optimized embeddings would be misleading.
>
> As shown in Tab. 3, although the optimized prototype (NCM) slightly outperforms class-mean NCM, both NCM variants consistently underperform the FC classifier—particularly in challenging scenarios like zero-base 10 phases. This highlights the advantage of the learnable FC classifier in capturing complex, non-Gaussian, and anisotropic decision boundaries that are crucial for robust continual learning. Meanwhile, in scenarios with less drift and more initial knowledge (e.g., half-base 10 phases), NCM performance is close to that of the FC classifier.

---

### Review · Reviewer_8KDr · 2025-05-28

**Summary Of Contributions:**

This paper presents YoooP and its enhanced variant YoooP+, two exemplar-free methods for class-incremental learning that address the challenges of catastrophic forgetting without relying on stored data from previous tasks. YoooP introduces a prototype-based framework that optimizes class representations through a mini-batch attentional mean-shift mechanism, producing compact and discriminative prototypes. YoooP+ further improves performance by generating synthetic data in angular space, preserving the original distribution of class features using a rotation-based augmentation strategy. The authors demonstrate that this approach results in higher-quality replay samples compared to prior methods that rely on Gaussian noise. Through comprehensive experiments on CIFAR-100 and TinyImageNet under both zero-base and half-base settings, the paper shows that YoooP and YoooP+ consistently outperform existing non-exemplar methods in terms of accuracy and forgetting. The work is further supported by ablation studies and analysis of representation quality, validating the effectiveness of the proposed components.

**Audience:**

Yes

**Broader Impact Concerns:**

No broader impact concerns

**Claims And Evidence:**

Yes

**Requested Changes:**

- Include runtime/memory comparisons with key baselines  to quantify the computational trade-offs introduced by prototype optimization and model interpolation.
- Clarify implementation details for prototype optimization and synthetic data replay, particularly the cost and frequency of these operations during training.

**Strengths And Weaknesses:**

Strengths:
- YoooP and YoooP+ outperform strong baselines across multiple benchmarks and learning settings, showing consistent gains in AIA and AA.
- The methods perform competitively despite not storing any real data, which is important for privacy-aware or resource-constrained applications.
- Unlike previous works that use Gaussian noise, YoooP+ generates data in angular space, leading to more realistic and diverse synthetic samples.

Weakness:
- Evaluation is limited to CIFAR-100 and TinyImageNet. Testing on larger datasets (if possible) would strengthen generalizability claims.
- While the paper introduces several accuracy-related metrics to evaluate performance, it does not provide comparisons in terms of memory usage or computational cost.

---

> ### Author Response · Authors · 2025-06-10
>
> Q1: Testing on larger datasets (if possible) would strengthen generalizability claims.
> ------------------
> **A1:** We appreciate the reviewer’s suggestion. In addition to CIFAR-100 and TinyImageNet, we evaluated our methods on **a larger Sub-ImageNet dataset (Appendix C, Table 5 and Figure 9)**. YoooP and YoooP+ consistently outperform non-exemplar-based baselines under both zero-base and half-base settings, confirming the robustness and generalizability of our approach.
>
>
> Q2: Include runtime/memory comparisons with key baselines to quantify the computational trade-offs introduced by prototype optimization and model interpolation.
> ------------------
> **A1:** Per your suggestion, we provide a detailed analysis of the computational cost and implementation details for prototype optimization and synthetic data replay in Tab. 1. As shown, our YoooP+ method achieves competitive or even lower MACs, GPU memory, and training time per epoch compared to other baselines (e.g., ADC, PASS, FeTrIL). Specifically, YoooP+ requires 571.26 G MACs, 9.86 GB GPU memory, and 22.07 seconds per epoch, demonstrating that prototype optimization and synthetic data replay introduce minimal overhead.
>
> Table 1: Efficiency on TinyImageNet under zero-base 10 phases setting, One GPU NVIDIA RTX A6000, BS=256, M=$10^6$, G=$10^9$, T=$10^{12}$.
> | Method | MACs↓ | Parameters↓ | GPU-Memo↓ | Time/Epoch↓ |
> |----------|-----------|-------------|-----------|-------------|
> | YoooP+ | 571.26 G | 11.17 M | 9.86 GB | 22.07 s |
> | YoooP | 571.26 G | 11.17 M | 9.53 GB | 17.10 s |
> | ADC | 571.26 G | 11.17M | 9.53 GB | 17.00 s |
> | PASS | 2.28 T | 11.18 M | >50 GB | 40.64 s |
> | FeTril | 571.26 G | 11.17 M | 10.72 GB | 349.15 s |
> | SSRE | 633.31 G | 11.17 M | 10.23 GB | 63.58 s |
> | LwF | 571.26 G | 11.17 M | 9.53 GB | 16.90 s |
> | IL2A | 2.74 T | 11.17 M | >50 GB | 97.4 s |

---

### Decision · Action_Editor_sVmH · 2025-06-30

**Recommendation:** Accept as is

**Audience:**

Yes

**Audience Explanation:**

The paper deals with the challenging continual learning scenario of class-incremental learning with the "exemplar free" requirement. It can be of interest to the community working on these topics, as evidenced by the large number of (very recent) comparisons requested to the authors.

**Claims And Evidence:**

Yes

**Claims Explanation:**

The paper proposes a technique to incrementally build class prototypes in an exemplar-free, class-incremental learning setting. The basic idea is to use a weighted average of the network embeddings to move the prototypes in a region of high-data density. They also introduce a way to sample data in this space via a rotation-based augmentation. The complete algorithm requires some additional loss and regularization terms.

We received three reviews which are currently split on their evaluation (accept, leaning accept, leaning reject). The first reviewer (8KDr) had two simple comments that were fully addressed in the rebuttal. Reviewer 7TYA was concerned about the lack of comparisons and discussion with Gaussian-based prototype methods, but they are satisfied by the additions during the rebuttal. Reviewer obox requested some additional comparisons, a discussion of the runtime, and an ablation study. As the authors have added all that was requested, I am unclear about their final evaluation, which seems to have skipped over the rebuttal (during which they did not interact further).

There is a consensus that the method, despite potentially missing comparisons, is sound. The experimental part is detailed, with many methods being compared, several ablations, and an analysis of all relevant aspects. On this issue, I believe the criteria of "accurate and clear evidence" are fully satisfied. Thus, I recommend acceptance of the paper.